# Biological fermentation pilot-scale systems and evaluation for commercial viability towards sustainable biohydrogen production

Quanguo Zhang[1,2], Youzhou Jiao[1], Chao He[1], Roger Ruan [3], Jianjun Hu[1], Jingzheng Ren [4], Sara Toniolo[5], Danping Jiang[1,2], Chaoyang Lu[1], Yameng Li [1,2] ✉, Yi Man[4], Huan Zhang [1,6] ✉, Zhiping Zhang [1,6] ✉, Chenxi Xia[2], Yi Wang[6], Yanyan Jing[1,6], Xueting Zhang[2], Ruojue Lin[4], Gang Li[6], Jianzhi Yue[1] & Nadeem Tahir [6]

Featuring high caloric value, clean-burning, and renewability, hydrogen is a fuel believed to be able to change energy structure worldwide. Biohydrogen production technologies effectively utilize waste biomass resources and produce high-purity hydrogen. Improvements have been made in the biohydrogen production process in recent years. However, there is a lack of operational data and sustainability analysis from pilot plants to provide a reference for commercial operations. In this report, based on spectrum coupling, thermal effect, and multiphase flow properties of hydrogen production, continuous pilot-scale biohydrogen production systems (dark and photo-fermentation) are established as a research subject. Then, pilot-scale hydrogen production systems are assessed in terms of sustainability. The system being evaluated, consumes 171,530 MJ of energy and emits 9.37 t of $CO_2$ eq when producing 1 t $H_2$, and has a payback period of 6.86 years. Our analysis also suggests future pathways towards effective biohydrogen production technology development and real-world implementation.

Finite supplies, and combustion-induced pollution issues, regarding the use of fossil fuels, have encouraged people to develop clean and renewable energy resources[1,2]. With high energy density and pollution-free combustion, hydrogen is perceived as the ideal alternative fuel[3–5]. Conventional physiochemical hydrogen production methods include coal gasification, steam methane reforming, propane steam reforming, solar photo-catalysis, etc.[6–9]. Most of these conventional approaches

consume non-renewable resources as raw materials, which not only increases the hydrogen production costs but also exacerbates environmental issues due to emissions of harmful pollutants. At present, hydrogen is given different color shades (black, gray, brown, gray, and green) according to the energy source used and its effects on the environment during hydrogen production technology[10]. Compared with conventional hydrogen production, the biohydrogen production

[1]Key Laboratory of New Materials and Facilities for Rural Renewable Energy of Ministry of Agriculture and Rural Affairs of China, College of Mechanical & Electrical engineering, Henan Agricultural University, Zhengzhou 450002, China. [2]Institute of Agricultural Engineering, Huanghe S & T University, Zhengzhou 450006, China. [3]Biorefining Center, University of Minnesota, Minneapolis and St. Paul, MN 55455, USA. [4]Department of Industrial and Systems Engineering, The Hong Kong Polytechnic University, Hong Kong, China. [5]Department of Management, University of Verona, via Cantarane 24, 37129 Verona, Italy. [6]Henan International Joint Laboratory of Biomass Energy and Nanomaterials, Collaborative Innovation Center of Biomass Energy, Henan Agricultural University, Zhengzhou 450002, China. ✉e-mail: liyameng2017@163.com; zhanghuan5754@163.com; zhangzhiping715@163.com

method utilizes microorganisms, resulting in fermentation or photosynthesis, to release free hydrogen gas[11–13], which is a more promising approach for preparing green hydrogen due to gentle processing technique and abundant reproducible raw materials[14–17]. Biological approaches include biophotolysis (direct and indirect), electrochemical systems (microbial electrolysis cells), and biological fermentation (dark fermentation, photo fermentation, and dark and photo fermentation)[18–22]. Among them, biological fermentation with the advantages of wider range of available raw materials (crop straw, livestock manure, kitchen waste) and potentially lower production costs has become a research focus.

Dark fermentation hydrogen production is the process in which bacteria convert organic matter into hydrogen and volatile fatty acids in the absence of light[23,24]. This process is characterized by a high hydrogen production rate and short processing times compared to photo-fermentation hydrogen production[23]. However, a large amount of volatile fatty acids remain in the fermentation effluent, resulting in low substrate conversion efficiency. In comparison, during photo-fermentation hydrogen production, photosynthetic bacteria convert monosaccharides and some volatile fatty acids into hydrogen[25,26], under conditions exposed to light. This process features high substrate conversion and mild reaction temperatures.

Combining the two mechanisms described above, dark fermentation can be used, first, for hydrogen production, and the remaining substrate can be fed to photosynthetic bacteria, thus creating a two-step "dark and photo fermentation", to improve the overall substrate conversion efficiency[26–29] Dark and photo-fermentation technology, which shows higher substrate conversion efficiency and shorter hydrogen production time, has attracted more and more people. Some measures have been adopted to promote hydrogen yield from organic matter, such as screening of hydrogen-producing bacteria, temperature regulation, optimization of reactors, addition of catalyst[22,30], etc., which were conducted in the lab. But there is a lack of data on large-scale hydrogen production systems. A pilot-scale bio-hydrogen production test is necessary prior to a commercial-scale application to analyze stable continuous operation feasibility and operation economic feasibility. Current research on these aforementioned technologies primarily relies on a kinetic model, and/or experimental data, to evaluate their industrial feasibility[31–34]. Questions, however, remain on how to minimize the error for scale-up modeling, and to improve reactor flexibility and ease of operation. So, before further scale-up of these biohydrogen production technologies can be achieved, more practical process parameters and feasibility evaluation studies need to be done. A better understanding of the scale-up modeling would significantly improve the process design scale-up decision-making.

In this article, a pilot-scale dark and photo-fermentation biological hydrogen production device was constructed. Biohydrogen production is a complex biochemical reaction. The hydrogen production performance is closely relationship to light source and density, temperature fluctuations, and rheological properties significantly affecting the energy flow and mutual transmission of the system. Light provides the necessary photoelectrons for photosynthetic bacteria to generate adenosine triphosphate, but not all wavelengths are suitable for the growth of photosynthetic bacteria and hydrogen production, the absorption of ineffective light can cause the "light saturation effect" resulting in lower hydrogen production performance. The biohydrogen production process is carried out with the participation of multiple enzymes in hydrogen production bacteria, which can be considered a temperature-sensitive biochemical reaction process due to the enzyme activity being significantly affected by temperature. For a fermentation system, the temperature fluctuation is caused by radiation heat transfer of light, biological heat, and external environment. Therefore, the study of lighting supply strategy and temperature fluctuation is crucial in improving hydrogen production performance.

In addition, the focus on the rheological characteristics of the fermentation system can achieve uniformity and stability of the system, improving the continuous hydrogen production capacity of the hydrogen production system.

Before the construction of the reactor, combining classical thermodynamic analysis methods with modern computational fluid dynamics methods, the distribution and transmission characteristics of light, heat, and mass of the fermentation system is analyzed, and based on these, the design of the dark and photo-fermentation biological hydrogen production is completed. As follows, first, a light supply system for the hydrogen production unit of photosynthetic fermentation was constructed, based on the spectral absorption characteristics of photosynthetic bacteria. Second, the temperature fluctuation of the fermentation system during the biological hydrogen production process was analyzed, providing a reference for the design of the temperature control system of the device. And then, the rheological characteristics of the fermentation broth in the reactor, as well as the temperature and concentration fields of the hydrogen production system, were simulated and analyzed, providing a reference for optimizing the reactor structure (Supplementary Fig. 1 shows the Research roadmap). Finally, we conducted a life cycle assessment (LCA) of a large-scale dark and photo-fermentation biological hydrogen production device and focused on the system energy consumption, environmental impact, and economic analysis. The hope is to provide a reference for the commercial operation of a biological hydrogen production system.

## Results
### Theoretical basis
**The principle of biological hydrogen production.** When carbohydrate is used as the substrate, dark fermentation bacteria convert the sugar into hydrogen and small-molecular acids through several pathways, such as the acetic acid route, the butyric acid route, and the ethyl alcohol route, the specific metabolic pathway is related to the hydrogen-producing bacteria used[35]. The small-molecular acids finally stay in the fermentation liquid. However, since dark fermentation bacteria cannot use small-molecular acids as a carbon source to produce hydrogen, the hydrogen yield from dark fermentation remains low. Illuminated by sunlight, the photo-fermentation bacteria can turn the small-molecular acids and sugar into hydrogen through inherent photosynthesis. Based on the hydrogen production mechanisms of dark fermentation and photo fermentation, the fermentation liquid, which leaves the dark fermentation, can be adopted as the substrate for photo fermentation. The combination of these two fermentation approaches can help improve the conversion efficiency of the substrate (Supplementary Fig. 2 shows the biochemical process of biological hydrogen production).

**Spectrum coupling.** Light is the energy source for the growth and hydrogen generation of photosynthetic bacteria as well as a basic premise for their survival. Sunlight is adopted as the primary light source for fermentation by photosynthetic bacteria in this work. Each species of photosynthetic bacteria has a unique absorption spectrum, which depends on the type and amount of photo-fermentation pigments it contains, and each pigment has its specific absorption peaks. As shown in Fig. 1a the absorption spectrum characteristics of photosynthetic bacteria vary with the wavelength of solar radiation. The photosynthetic bacteria have obvious absorption peaks at 325 nm, 382 nm, 490 nm, 590 nm and 807 nm, 865 nm in the infrared light region, and photosynthetic bacteria can utilize these spectra (The reaction device is shown in Supplementary Fig. 3).

Then, the filters are used to screen the wavelengths suitable for the growth of photosynthetic bacteria and hydrogen production to eliminate the "light saturation effect" caused by the absorption of excessive light energy and obtain the coupling between

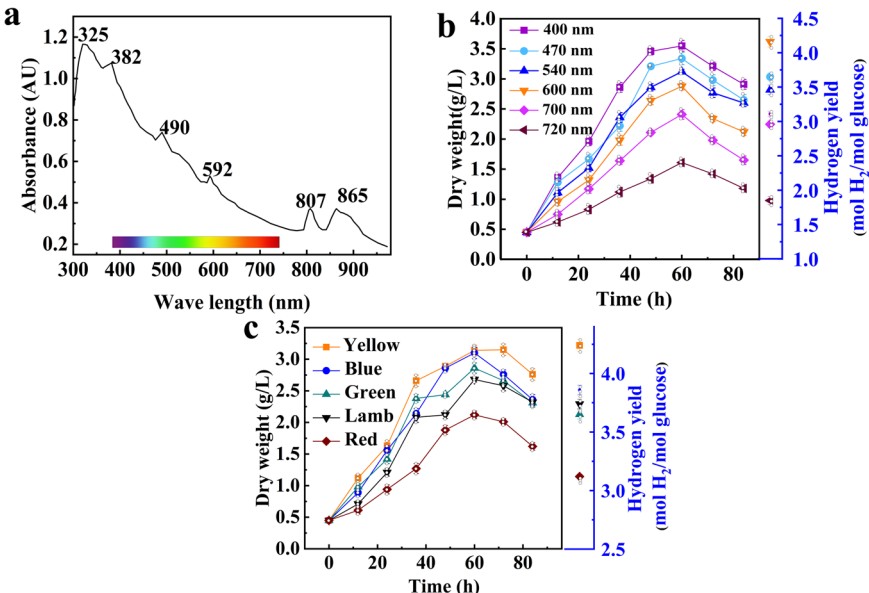

**Fig. 1 | Analysis of light absorption and growth characteristics of photosynthetic hydrogen-producing bacteria. a** Characteristics of the absorption of photosynthetic bacteria. **b** photosynthetic bacteria growth and hydrogen production under different wavelengths ($n = 3$, independent experiments). **c** Photosynthetic bacteria growth and hydrogen production under different light sources ($n = 3$, independent experiments).

photosynthetic bacteria and solar energy. The growth characteristics and hydrogen production effect of photosynthetic bacteria under different wavelengths are shown in Fig. 1b. The growth effect of photosynthetic bacteria is best under the 400 nm spectrum; the maximum cell dry weight can reach 3.55 g/L with the minimum hydrogen production. The maximum hydrogen production occurs at a wavelength of 600 nm, which is 4.16 mol $H_2$/mol glucose. Therefore, the wavelength suitable for the growth of photosynthetic bacteria does not lead to a consistent hydrogen production effect. It appears that the activities of enzymes involved in hydrogen production occur at different wavelengths. Thus, light sources of different colors are selected to replace sunlight to ensure the efficient performance of the experiments in the absence of sunlight. The growth and hydrogen production characteristics of photosynthetic bacteria under different light sources (light-emitting-diodes (LEDs) purchased from Cree Shanghai Opto Development Limited) are shown in Fig. 1c. Photosynthetic bacteria show better growth and hydrogen production characteristics under yellow and blue light; the corresponding hydrogen production is 4.24 mol $H_2$/mol glucose and 3.74 mol $H_2$/mol glucose, respectively, which is higher than that of white light sources (LED). The reason might be due to the light emitted by white light sources is a composite light composed of multiple monochromatic lights, and higher radiation energy is absorbed by photosynthetic bacteria, resulting in the "light saturation effect" that reduces the activity of photosynthetic bacteria.

The aforementioned experimental design is illustrated in the supporting information.

**Thermal effect**. During the hydrogen production by photosynthetic bacteria, light irradiates the reaction system through the reactor. Part of the physical factors accepted by photosynthetic bacteria is converted into thermal energy and released directly, and part of the physical factors is absorbed by the biological system to increase the cell's metabolism with the released thermal energy. For dark fermentation, only the latter occurs. The biological effects caused by the above impact are the thermal effect of hydrogen production of fermentation bacteria. The generated heat leads to increased temperature, which affects the strength of the enzyme activity, and then affects the hydrogen production efficiency. The experiments are designed to analyze the thermal effect of different initial temperatures and light

intensities on hydrogen production, to provide a reference for the design of temperature control systems in pilot-scale reactors. For thermal effect experiments, the temperature-varying system experiment is carried out in a vacuum reaction bottle, the constant temperature system experiment is performed in regular glass bottles maintained at a constant temperature realized by the temperature control system. The reaction system involves several factors, such as the biochemical reaction, light source, and temperature control. (Supplementary Fig. 4 shows the heat energy transmission of the system. Supplementary Fig. 5 shows an illustration of the experiment equipment of the hydrogen production system on thermal effect.)

The temperature fluctuation of the temperature-varying system at different initial temperatures for the dark fermentation system and photo-fermentation system is shown in Fig. 2a, d. During the first 12 h of fermentation, the system temperature at different initial temperatures increases to a large extent. The system temperature rises slowly from 12 to 20 h, and remains basically stable after 20 h. The variation rate of the temperature does not show regularity with the change of the initial temperature for the fermentation system, but the variation rate of the temperature for photo fermentation is greater than that of dark fermentation, which may be caused by light radiation. The maximum temperature increment is 3.43 °C, detected in the photo-fermentation system with an initial temperature of 27 °C. The magnitude of the temperature change is 36 °C, 33 °C, 39 °C, and 30 °C for dark fermentation and is 27 °C, 30 °C, 24 °C, and 33 °C for photo fermentation. The heat production rate at different initial temperatures is shown in Fig. 2b, e. The heat production rate increases rapidly from 2 to 6 h for the dark fermentation system and then decreases until the system reaches heat balance, but for the photo-fermentation system, the heat production rate increases rapidly from 2 to 8 h, and then decreases. The reason can be explained by the fact that dark fermentation bacteria can quickly adapt to a fermentation environment to grow and produce hydrogen. The maximum heat generation rate (1.14 kJ/L·h) is detected in the photo-fermentation system with an initial temperature of 27 °C. Figure 2c, f shows the hydrogen production at different initial temperatures. At 30 °C and 33 °C in the dark fermentation system, 24 and 27 °C in the photo-fermentation system, the hydrogen production is affected by temperature fluctuations and is greater than that of the constant temperature system at the same

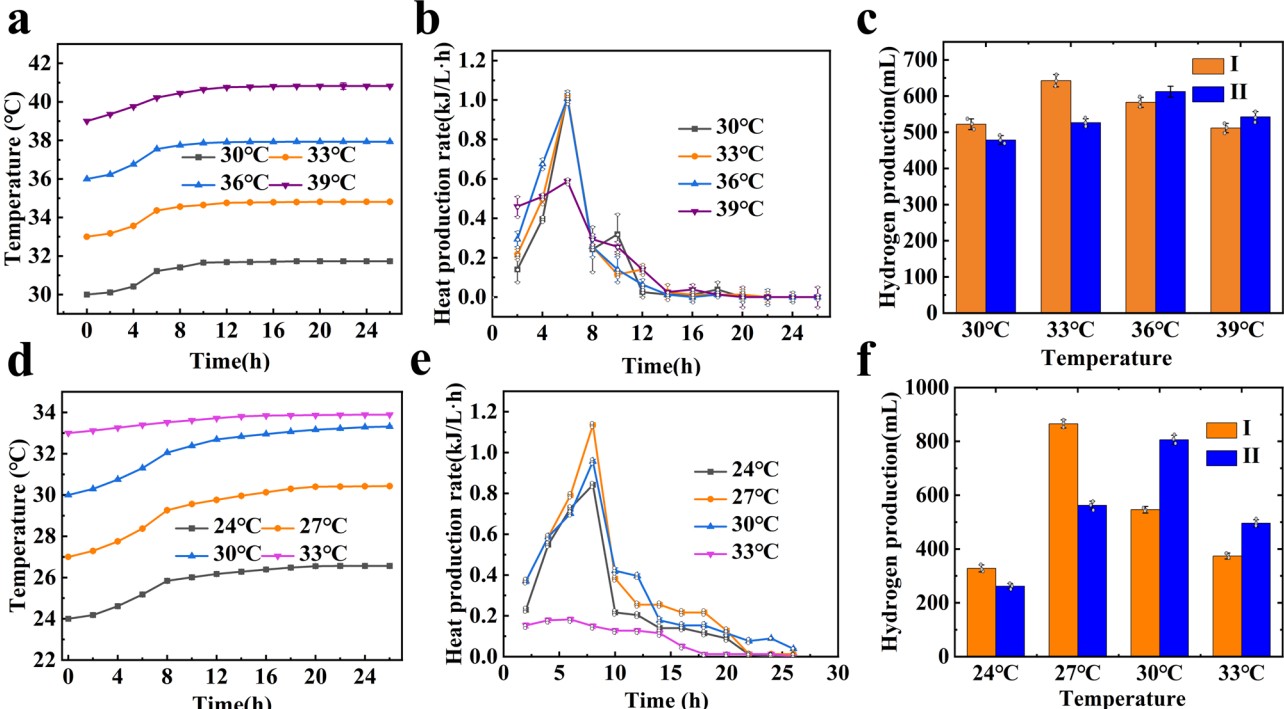

**Fig. 2 | Effects of operation temperature on hydrogen production system.**
Effects of operation temperature on hydrogen production system. Dark fermentation: **a** Temperature fluctuation of the system at different initial temperatures. **b** Heat production rate at different initial temperatures ($n=3$, independent experiments). **c** Hydrogen production at different initial temperatures ($n=3$, independent experiments). Photo fermentation: **d** Temperature fluctuation of the system at different initial temperatures. **e** Heat production rate at different initial temperatures ($n=3$, independent experiments). **f** Hydrogen production at different initial temperatures ($n=3$, independent experiments). (I: Temperature-varying system. II: Constant temperature system).

initial temperature. The opposite situation occurs when the initial temperature is 36 and 39 °C in the dark fermentation system, 30 and 33 °C in the photo-fermentation system. The reason may be due to the activity of the enzyme is negatively affected by the temperature increase. The maximum hydrogen production occurs at 27 °C while the minimum occurs at 24 °C for the photo-fermentation system.

Table 1 illustrates the thermal effect on dinitrogenase (photo-fermentation bacteria) and hydrogenase (dark fermentation bacteria) activity at different initial temperatures. For photo-fermentation hydrogen production, at the initial temperature of 24 °C and 27 °C, with the heat accumulation, the dinitrogenase in the temperature-varying system is higher than that of the constant temperature system, which is opposite to the situations at 30 °C and 33 °C. With the change of initial temperature, the activities of enzymes increase and then decrease, which is consistent with the hydrogen production ability of photosynthetic bacteria in temperature-varying systems. During the dark fermentation process, the change trend in hydrogenase activity is the same as that of dinitrogenase of photosynthetic bacteria, the maximum hydrogenase activity was detected at the initial temperature of 33 °C in a temperature-varying system.

Photosynthetic bacteria require appropriate light intensity for hydrogen production. The light intensity affects the number of photons captured by the photosynthetic bacteria, the formation of ATP (Adenosine Triphosphate), and the proton gradient, and plays an important role in the hydrogen production of photosynthetic bacteria. The effects of different light intensities on the system temperature are shown in Fig. 3, the system temperature rises with increasing light intensity. When the light intensity is 500 Lx, the temperature fluctuation is the smallest, and the equilibrium temperature is 28.57 °C. The heat production rate at different light intensities is shown in Fig. 3b. The maximum heat generation rate occurred at 8 h for every system. When the light intensity is 3000 Lx, the maximum heat generation rate of the system is 1.32 kJ/(L·h). The light intensity, system temperature, and accumulated heat are consistent with the maximum heat production rate. Figure 3c shows the thermal effect on hydrogen production at different light intensities. The hydrogen production of a temperature-varying system is greater than that of a constant temperature system. The maximum hydrogen production occurs at the light intensity of 3000 Lx, the values are 894 mL and 652 mL for the temperature-varying system and constant temperature system.

Table 2 illustrates the results of the thermal effect on dinitrogenase activity at different light intensities. The enzyme activities with heat accumulation are higher than those without heat

## Table.1 | The effect of operation temperature on enzyme activities

| | Temperature (°C) | Dinitrogenase activity (nmol $C_2H_4$/mL bacterial solution/h) | |
| --- | --- | --- | --- |
| | | I | II |
| Photo-fermentation system | 24 | 102 | 65 |
| | 27 | 520 | 405 |
| | 30 | 430 | 446 |
| | 33 | 289 | 328 |
| | | Hydrogenase activity (nmol $H_2$/mL bacterial solution/h) | |
| | | I | II |
| Dark fermentation system | 30 | 305 | 232 |
| | 33 | 685 | 512 |
| | 36 | 562 | 623 |
| | 39 | 423 | 506 |

(I: Temperature-varying system. II: Constant temperature system)

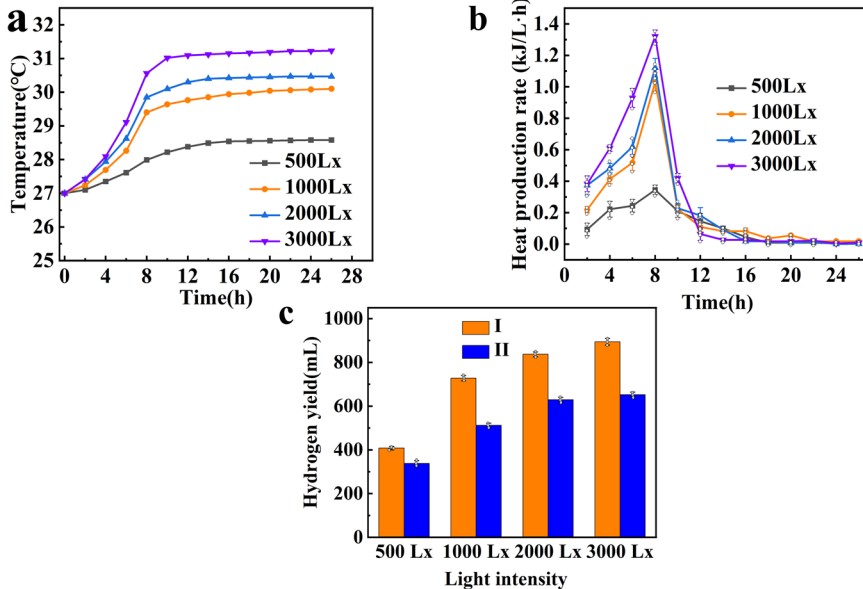

**Fig. 3 | Effects of light intensities on hydrogen production system.** Effects of light intensities on photo-fermentation hydrogen production system. **a** Temperature fluctuation of the system at different light intensities. **b** Heat production rate at different light intensities ($n = 3$, independent experiments). **c** Hydrogen production at different light intensities ($n = 3$, independent experiments).

accumulation. The enzyme activities rise with the increase of light intensity, which is consistent with the hydrogen production ability of photosynthetic bacteria, which benefit from the thermal effect. Increasing the light intensity in a certain range is conducive to the improvement of the enzyme activities, and thus to the improvement of hydrogen production capacity.

## Liquid rheological properties of baffle plate reactor and multiphase flow

The liquid rheological properties and flow behavior are the main characteristics of hydrogen production. In view of the complexity of the fluid itself and the reaction process, this research simulated the velocity field and concentration field distribution, of a continuous hydrogen production system, providing a reference for optimizing the reactor structure. A mathematical model of the multiphase flow field was established, and the velocity, and concentration of material in the liquid were explored, to remove and reduce the liquid retention zone and liquid flow shock, and prolong the service life of the reactor. The rheological properties and simulation methods of this theory are presented in the supporting information. In the early stage of the reactor design, the baffle reactor is the most ideal reactor for continuous fermentation of hydrogen production. Mixing was accomplished by the flow of the fermentation media through the baffles in the bioreactors. Therefore, the baffle plate reactor was used in subsequent experiments. (The analytical methods were explained in the Supplementary Methods. The grid of the reacting region is shown in Supplementary Fig. 6)

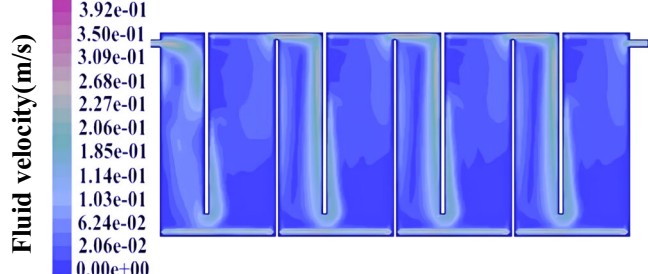

**Fig. 4 | Characteristics of velocity distribution of mixture in reactor.** Simulation of the velocity distribution of mixture in a baffled biohydrogen production reactor.

The velocity field distribution of the mixture in the baffle plate reactor is shown in Fig. 4. The velocity of the inlet mixture is relatively high; after entering the reactor, the velocity of the fluid decreases rapidly due to the larger diameter. The fluid velocity tends to increase due to the sudden decrease in the diameter at the bottom of the baffle. The velocity of the mixture in the down-flow chamber is higher than the velocity in the up-flow chamber. There is a vortex at each baffle position, and the maximum velocity of the reaction system is 0.412 m/s. Each compartment has a detention zone and the area with a lower velocity in the entire reactor account for almost 1/6th. The reactor design should minimize the detention zone, and enhance the stirring effect to improve the efficiency of the hydrogen production, which can be achieved by improving the reactor design or increasing the feed rate.

The concentration field distribution of the mixture is shown in Fig. 5, and the situation of the liquid phase is shown in Fig. 5a. From the height of the mixture lifted by the fluid flow at the bottom, it can be concluded that the concentration of the liquid phase at the bottom of the reactor decreased from 97.0% at the beginning to 84.9%–87.9% in the yellow-red part. The concentration of the yellow part changes from 81.8% to 84.9%; the light green area has a concentration of 75.8% and occupies the largest area. The minimum value of the liquid phase concentration is 39.6%, which appeared in the solid precipitate at the up-flow chamber. The above conditions are consistent with the actual operation of the reactor. The solid-phase concentration distribution is

## Table 2 | The effect of light intensities on enzyme activities

| Light intensities (Lx) | Dinitrogenase activity (nmol $C_2H_4$/mL bacterial solution/h) | |
|---|---|---|
| | I | II |
| 500 | 170 | 89 |
| 1000 | 320 | 228 |
| 2000 | 495 | 370 |
| 3000 | 516 | 390 |

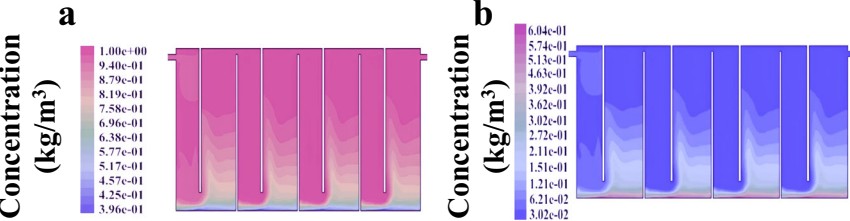

**Fig. 5 | Simulation analysis of solid and liquid phase concentration distribution in baffled plate biohydrogen production reactor. a** Liquid phase concentration field distribution. **b** Solid-phase concentration field distribution.

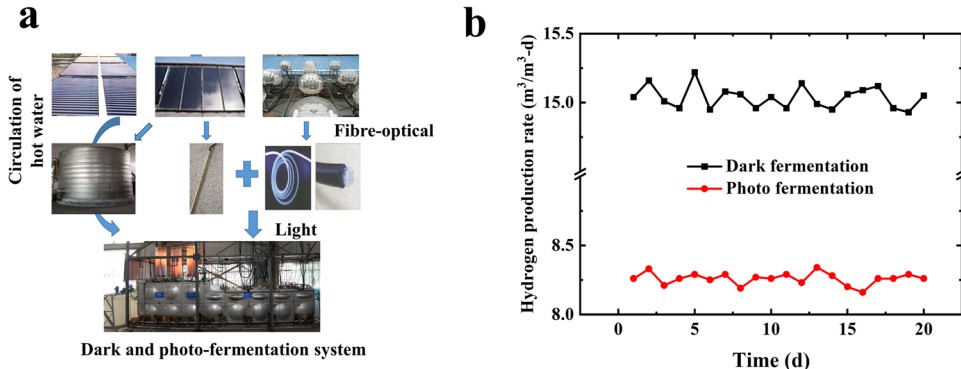

**Fig. 6 | Composition and operational analysis of dark and photo-fermentation hydrogen production reactor system. a** Real-world operation system. **b** hydrogen production rate of the fermentation system.

shown in Fig. 5b. The concentration distribution of the solid phase is exactly the opposite of the concentration distribution of the liquid phase. The solid-phase concentration increases from the upper part of the reactor to the bottom, and there is little difference in the solid-phase concentration at the same position in each compartment. The pushing movement causes the precipitated solid particles to move forward, and most of them are collected in the upstream chamber. The height and concentration of solid-phase distribution in the upstream chamber are larger than that in the downstream chamber, which is opposite to the liquid phase. These experiments show that there is an obvious concentration gradient and mass transfer in the baffled hydrogen production system. The fluid flow is conducive to the uniform distribution of concentration and mass. As a result of the baffles, the stirring effect of the reactor is enhanced, which achieves the function of automatic feed mixing and reduces energy waste.

## Pilot-scale experimental device

A pilot-scale baffled bioreactor for sequential dark and photo-fermentation continuous hydrogen production was established, based on previous research results[36,37]. The structural dimensions were described in detail in previous articles[36,38,39] (Supplementary Fig. 7. shows the system structure diagram of pilot-scale baffled continuous flow dark and photo-fermentation hydrogen production reactor). During dark fermentation, hydrogen generation substrate (pH = 5.5) and dark fermentation bacteria are injected at a fixed ratio of 4:1 from the rightmost tank, by a peristaltic pump. When the liquid level in the first chamber is higher than the inter-chamber baffle, the liquid will flow into the second chamber, and the third chamber, in the same way. When dark fermentation is carried out alone, the liquid fermentation waste will be discharged from the liquid outlet after fermentation in the tank. For the hydrogen production system, a mixing chamber and a dark fermentation broth treatment chamber are located between the dark fermentation unit and the photo-fermentation unit. By regulating the flow valve, the fermentation liquid from the dark fermentation is diverted from the liquid outlet to the dark fermentation broth treatment chamber, where the liquid undergoes zeolite treatment

(removing excessive ammonium ions) and ultraviolet sterilization. It then goes into the mixing chamber to be mixed with photo-fermentation substrate, and regulated in pH value (pH = 7). After that, the fermentation liquid enters the photo-fermentation tank through the peristaltic pump at a ratio of 4:1 to the photo-fermentation bacteria. After the fermentation in the last chamber, the fermentation liquid is discharged from the liquid outlet. Every tank is provided with an ascending cubicle and a descending cubicle. The ascending cubicle acts as the major photoreaction area. To ensure enough area for photoreaction, the volume of the ascending cubicle is four times larger than that of the descending one; to meet the sunlight demand during photo-fermentation hydrogen production, a light source is placed within a circular glass tube, and then into the fermentation liquid. During the daytime, sunlight is transmitted into the reactor via the solar tracer and light concentrator to provide light; however, on cloudy, rainy days or night time, the lighting comes from the LED lamp (composition of yellow LED and blue LED). Electricity required for the operation of the reaction device is supplied by a storage battery, which saves the energy generated by solar panels[36,39]. In the periphery of the reactor, an insulating layer is provided. The hydrogen production system depends on circulating hot water from the solar water heater, to maintain its operating temperature. In order to reduce heat loss around the reactor, insulating material is stuffed into the periphery of the device.

Circulating hot water maintains the reaction temperature. The real-world operating system of the experimental equipment is shown in Fig. 6a (Supplementary Fig. 8. shows the 3D view of the reactors).

During fermentation, the generated gas enters the gas tank through the gas pipe

Enzymatic hydrolysate of corn straw(reducing sugar) is used, at a concentration of 25 g/L, as the substrate for hydrogen production via the dark and photo-fermentation hydrogen production device. First, the reactor was operated in batch mode for 30 d to enrich the functional strains, and then the system was switched to a continuous model. After two weeks of operation, the system tends to stabilize. During the two-year operation period, the operating process was

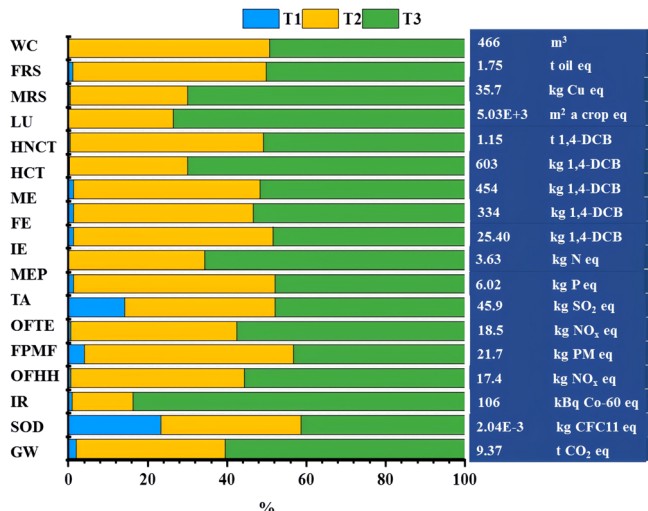

**Fig. 7 | Life cycle environmental impact of hydrogen production of dark and photo fermentation.** T1: Straw pulverization; T2: Enzymolysis; T3: Hydrogen production.

optimized[36,38,39]. Once we have collected data for 20 days of stable operation, as shown in Fig. 7b, the average hydrogen production rate of the dark fermentation unit is 15.04 m³/m³·d, and the average hydrogen production rate of the photosynthetic fermentation is 8.26 m³/m³·d. Through calculation, it can be found that the device can produce 10 kg of hydrogen per day. The average hydrogen production rate obtained in the paper was higher than that of semi-pilot-scale up-flow anaerobic sludge blanket reactor[40] and packed bed biofilm reactor[41]. For the pilot-scale hydrogen production system in the paper, each chamber is a baffled reactor and can independently ferment for hydrogen production, which shows good flexibility and scalability to meet the construction of fermentation systems of different scales. For a continuous hydrogen production system, the stable and continuous generation of hydrogen is an important goal, which may be achieved under a high organic loading rate, leading to the loss of residual hydrogen-producing bacteria and a certain amount of organic matter in the tail liquid after fermentation. Therefore, some technologies should be adopted to treat the hydrogen production tail liquid to achieve efficient utilization of substrates, such as the preparation of liquid organic fertilizer, and methane production from hydrogen production tail liquid. (See the supporting information for the experimental process and other operational data.)

## Energy consumption analysis

In the process of hydrogen production by fermentation, energy consumption can be divided into three stages: (1) pulverization stage, (2) enzymolysis stage, and (3) fermentation stage.

During the pulverization stage, the energy consumption comes primarily from the raw material crusher. For the enzymolysis stage, the energy required includes both heat to maintain the hydrolysis temperature and mechanical energy for the agitator operation. During the fermentation stage, energy consumption mainly comes from feeding devices, lighting devices, mixing devices, and online monitoring systems. Results show that the dark and photo-fermentation system consumes 171,530 MJ to generate 1 t hydrogen (171.53 MJ/kg H₂), this is supplied by solar energy, which is close to the energy consumption of green hydrogen production by electrolysis water using solar and wind energy (183.58 MJ/kg H₂[42], 150.10 MJ/kg H₂[43], 168.82 MJ/kg H₂[43]). But the energy consumption is lower than that of hydrogen production via using deep in-situ gasification-based coal-to-hydrogen (349.55 MJ/kg H₂)[44]. During the operation, energy consumption mainly occurs in the enzymolysis stage at 53.04%, the pulverization stage at 9.76%, and the

fermentation stage at 37.2%. In the future, research on high-activity cellulase and high solid-phase enzyme hydrolysis technology should be followed to reduce to reduce the energy consumption of enzyme hydrolysis processes.

Energy consumption is one resulting aspect of the environment from the hydrogen production system. To analyze the environmental impact of such systems in detail, it is necessary to perform a life cycle environmental assessment. The following sections will elaborate on this assessment of the hydrogen production system.

## Environmental impact analysis

**Life cycle inventory.** According to ISO 14040 and ISO 14044[45], LCA for hydrogen production systems is made by using SimaPro 8.5based on ReCiPe2016 Midpoint method[46]. The life cycle data are collected mostly from China and supplemented with the database in Ecoivent 3.1 so that the assessment results can be more representative in China. ReCiPe 2016 Midpoint (H), adopted in the life cycle assessment, incorporates 18 categories: (1) Global warming (GW), (2) Stratospheric ozone depletion (SOD), (3) Ionizing radiation (IR), (4) Ozone formation, Human health (OFHH), (5) Fine particulate matter formation (FPMF), (6) Ozone formation, Terrestrial ecosystems (OFTE), (7) Terrestrial acidification (TA), (8) Freshwater eutrophication (FEP), (9) Marine eutrophication (MEP), (10) Terrestrial ecotoxicity (TE), (11) Freshwater ecotoxicity (FE), (12) Marine ecotoxicity (ME), (13) Human carcinogenic toxicity (HCT), (14) Human non-carcinogenic toxicity (HNCT), (15) Land use (LU), (16) Mineral resource scarcity (MRS), (17) Fossil resource scarcity (FRS), (18) Water consumption (WC).

The functional unit is 1 metric ton of H₂. The boundaries of the fermentation system are listed in Supplementary Fig. 10. A list of energy consumption from the straw pulverization to the ending of hydrogen production is listed in Supplementary Table 1.

Figure 7 demonstrates the life cycle environment impacts and their contribution to different life cycle stages of hydrogen production systems calculated by the ReCiPe 2016 Midpoint (H) method. During the life cycle of hydrogen production by dark and photo fermentation, the contribution to GW is 9.37 t CO₂ eq in producing 1 t hydrogen (9.37 kg CO₂ eq/kg H₂). The GW for the hydrogen production system, the GW was mainly originated from the hydrogen production process, was over 60% of the total amount. The reason may be due to the addition of chemical reagents during the fermentation stage. This provides a research direction for us to reduce GW from biological hydrogen production processes. Supplementary Table 2 shows the GW from different hydrogen production technologies. The GW of hydrogen production via biomass gasification and coal gasification is approximately 10.56 kg CO₂ eq/kg H₂[47] and 18 kg CO₂ eq/kg[48], respectively. The GW of green hydrogen production via water electrolysis using wind power or solar power shows a lower value varying from 9.4 to 0.3 kg CO₂ eq/kg[49,50]. But renewable power generation is limited by land area available for photovoltaic panels and/or wind turbines, when the grid electricity is used to compensate for insufficient wind or solar electricity, the GW could reach to 25.93 CO₂ eq/kg H₂[51]. Compared to green hydrogen production via water electrolysis, the biological fermentation hydrogen production system exhibits advantages in resource recycling and simultaneously achieving waste treatment and clean energy production. With the application of carbon capture and storage technology, the preparation process of materials gradually becomes cleaner, which will reduce the GW of the biological fermentation hydrogen production system.

Through literature comparison, it was also found that different biomass conversion technologies exhibit different emission reduction capabilities based on LCA[52–54], caused by the differences in assessment methods, functional units, and system boundaries[55], but the final results all indicate development and utilization of biomass helps to net-zero emissions.

**Table 3 | Capital investment and cost analysis**

| Initial investment (10⁴ CNY) | Raw materials (t) | Reagent consumption (CNY/t H₂) | Water (CNY/t H₂) | Construction cost (CNY/t H₂) | Maintenance cost (CNY/year) | Welfare (CNY/month) | Cost (CNY/t H₂) |
|---|---|---|---|---|---|---|---|
| 800, (109.33$) | 40 | 9697.6 (1325.35$) | 3520 (481.07$D) | 950 (129.83$) | 12,000 (1640.02$) | 3000 (410$) | 43,875.16 (5996.33$) |

**Sensitivity analysis.** Cellulase is an indispensable reagent for straw hydrolysis, which can hydrolyze large-molecule glucans into small-molecule organic saccharides. The activity of cellulase determines the dosage of cellulose and further affects the whole hydrogen production life cycle on the environmental impact. With advancements in science and technology, the activity of cellulase will be gradually improved. This study supposes the cellulase activity is raised by 15%, its dosage will be reduced by 15% under the same conditions, the scheme was defined as S1. The environmental impact is compared before and after the change. In addition, an improvement in cellulase activity can also alleviate the inhibiting effect of reducing sugar during enzymolysis. It can be found that citric acid and sodium citrate during enzymolysis have made a significant contribution to the environmental impact. Thus, another hypothesis is proposed here: based on improved cellulase activity, the solid-to-liquid ratio dosage of cellulase should be changed (from 1:10 to 1:8); this scheme was defined as S2 (The analysis process is shown in supplementary). By comparison, it can be found that the decrease in the solid-to-liquid ratio (S2) can minimize the environmental impact in the life cycle of hydrogen production systems due to the increase in solid-to-liquid ratio and decrease the consumption of substances. (See the supporting information for the experimental process and other operational data.)

**Life cycle costing assessment**
**Cost estimation.** The hydrogen production system is expected to work for 20 years in total, and 360 days per year. The depreciation rate of the fixed asset is 5%[52]; raw material purchasing price is determined by the local market; annual maintenance cost accounts for about 1.5% of the fixed assets[56]. The project is constructed in a tertiary administrative region in China, so the salary of the laborers refers to the lowest wage standard in such region. Based on market reagent price, water, and power charge, and other expenditures, the reagent expense for operating the experiments of the hydrogen production system can be calculated. For the designed dark and photo-fermentation system, cost of 43,875.16 CNY / t H₂ (5999.5 $/t or 5.6 $/kg). Compared with other reported results, the hydrogen obtained from the combined system shows better market competitiveness (Supplementary Table 3).

These economic estimates are offered in the Supplementary Note 6. Table 3 provides information about the annual profit and loss statement and dynamic economic analysis of the whole project.

When the (net present value) NPV > 0, it means the plan is economically feasible. Tp is the investment payback period, and IRR is the internal return of rate. When the IRR is higher than the benchmark discount rate (8%), the project has a favorable economic effect[31]. According to the market price at the project site, the sale price of hydrogen is set at 56 CHY/kg (7.65$/kg). Based on the data of Table 3, the financial net present value (NPV) of dark and photo-fermentation system is 584,400 CNY (79,843$/kg), the investment payback periods are estimated to be 6.86 years and IRR is 16.84% for dark and photo-fermentation system, the payback period is lower than 10.28 years for hydrogen production with water electrolysis (IRR 10.28%)[57]. (Supplementary Table 4)

## Discussion
The construction and operation of a large-scale biological hydrogen production system is a necessary step to achieve the commercialization of biological hydrogen production. In this article, a multi-spectral coupling light supply system, with solar energy as the energy source, was constructed, based on the light absorption characteristics of the

hydrogen-producing microorganisms. The biohydrogen production reactor was optimized according to the thermal effect and multiphase flow characteristics of the fermentation system. Based on the above theoretical foundations, we have constructed a pilot-scale biological hydrogen production reactor that relies entirely on clean energy operation. Evaluation and analysis results show when the system produces 1 t of H₂, it will consume 171,530 MJ of energy and emit greenhouse gas (GHG) 9.37 t CO₂ eq.

In the sensitivity analysis of the life cycle assessment of the system, it was found that increasing the solid-liquid ratio of the fermentation system can reduce greenhouse gas emissions during the hydrogen production process. The theoretical research and fermentation equipment design and operation in the paper may provide theoretical and equipment support for the development of biomass green hydrogen technology, and provide systematic solutions for carbon sequestration and emission reduction in the industrial and agricultural fields, supporting the development of carbon neutrality and circular economy.

## Methods
### Hydrogen production bacteria
Dark fermentation hydrogen production bacteria composed of *Enterococcus*, *Sporanaerobacter*, *Para clostridium*, and *Clostridium_sensu_stricto_1* provided by Key Laboratory of New Materials and Equipment for Rural Renewable Resources, Henan Agricultural University[58]. The bacteria used for photo-fermentation hydrogen production were HAU-M1 photosynthetic bacteria from the Key Laboratory of New Materials and Equipment for Rural Renewable Resources, Henan Agricultural University. The HAU-M1 was composed of oval-shaped *Rhodospirillum rubrum* (27%), rod-shaped *Rhodopseudomonas palustris* (28%), ellipsoid-shaped or short rod-shaped *Rhodopseudomonas capsulata* (25%), *Rhodobacter sphaeroides* (9%), and *Rhodobacter capsulatus* (11%)[59]. Dark fermentation hydrogen production bacteria and photo-fermentation hydrogen production bacteria were originally screened from sludge (municipal wastewater treatment plants), cow dung, pig manure, and camel dung[58,60]. First, the initial fluid was heat-treated at 100 °C for 10 min, and then, adding a specific culture medium to the heat-treated liquid to enrich the hydrogen-producing bacteria. The enrichment medium of dark fermentation hydrogen bacteria composed of soya peptone (5 g/L), pancreatic peptone (15 g/L) and NaCl (5 g/L), for photo-fermentation hydrogen production bacteria, enrichment medium contained 1 g/L of NH₄Cl, 2 g/L of NaHCO₃, 0.2 g/L of K₂HPO₄, 3 g/L of CH₃COONa, 0.2 g/L of MgSO₄·7H₂O, 2 g/L of NaCl, and 1 g/L of yeast extract. The augmented period was conducted for 8 cycles that lasted for almost 32d, then the hydrogen-producing bacteria were isolated and identified.

### Spectral absorption characteristics
The logarithmic hydrogen-producing bacteria are suspended in a 60% sucrose solution after centrifugation with 5500 rpm and washing, scanned with an Agilent8453 UV-VIS spectrophotometer (Agilent, USA) with a scanning range of 190-1100 nm to detect the absorption of light by photosynthetic bacteria[61].

### Experiment on the influence of spectra and light sources on hydrogen production
Using the light from the xenon lamp that is closest to the solar spectrum as the light source, a filter device is installed in front of the light

collector to obtain spectra of different wavelengths. The filters used in this article are 100 nm bandpass filters DTB400, DTB470, DTB540, DTB600, DTB700, and DTB720 (Nantong Yinxing Optics Co., Ltd, China) to process the collected xenon lamp light, in order to ensure the uniformity of photosynthetic bacteria receiving light and avoid losses caused by light scattering, the xenon light is collected by a collector and transmitted to the glass bottle by optical filters (the reaction device is shown in Supplementary Fig. 3a).

When conducting experiments with different light sources (LEDs), the light sources used include a yellow light source, blue light source, green light source, red light source, and white light source. To control a certain temperature, the reaction bottle is placed in a constant temperature box, and each group of experiments is conducted three times (the reaction device is shown in Supplementary Fig. 3b).

The test conditions are set as follows: 25% of the inoculation amount, 30 °C of temperature, 3000 Lx of light, pH 7, and 10 g/L of glucose concentration.

The dry cell weight of hydrogen-producing microorganisms in culture medium is measured using the following method: The bacterial solution is centrifuged at 10,000 rpm for 5 min, washed twice with distilled water, and then placed in a drying oven at 105 °C until a constant weight is achieved. The weight is then measured using an electronic balance.

The hydrogen concentration in the biogas is detected by gas chromatograph (Agilent, 6820 GC-14B). The light intensity is measured with a digital Lux meter (TES-1330A, Taiwan, China). The oxidation-reduction potential (ORP) of the reaction solution is measured by oxidation-reduction potentiometer (SX712, Shanghai, China). The pH of the solution is measured by a pH meter (PHS-3C, Shanghai, China)[26].

## Thermal effect analysis

During the hydrogen production by photosynthetic bacteria, light irradiates the reaction system through the reactor, part of the physical factors accepted by photosynthetic bacteria is converted into thermal and released directly, and part of the physical factors is absorbed by the biological system to increase the cell's metabolism with the released thermal. For dark fermentation, only the latter occurs. The biological effects caused by the above impact are the thermal effect of hydrogen production bacteria. The generated heat leads to the increased temperature, which affects the strength of the enzyme activity, and then affects the hydrogen production efficiency. The reaction system involves many factors, such as biochemical reaction, light source, temperature control, etc., the heat energy transmission of the system is shown in Supplementary Fig. 4.

The heat balance equation of the photo-fermentation system is shown in Eq. (1).

$$Q_g + Q_e = Q_r + Q_w + Q_j \tag{1}$$

For dark fermentation, the heat balance equation is shown in Eq. (2).

$$Q_e = Q_r \tag{2}$$

$Q_g$ – photothermal, $Q_r$ – heat from hydrogen production, $Q_w$ – heat from reflection or refraction of photosynthetic bacteria, $Q_j$ – heat from reflection or refraction of matrix, $Q_e$ – heat transfer between the reaction system and the environment.

## The temperature-varying system I

The reaction operation system is shown in Supplementary Fig. 4. The vacuum-insulated reaction bottle with a reaction volume of 700 mL is used as the reaction vessel, SWJ-I c precision digital thermometer, with a temperature measurement range of −20-100 °C and a resolution of 0.01 °C, is adopted. The prepared reaction bottle is placed in a modified constant temperature box, and the control system of the constant

temperature box is connected to the computer. When the temperature sensor senses a change in the temperature of the vacuum reaction bottle, the signal is transmitted and displayed on the precision digital thermometer, and further transmitted to the computer. The temperature change signal is then fed back to the constant temperature box through the set program. The incubator utilizes its own control ability to adjust the temperature to ensure that the temperature inside the incubator is consistent with the temperature inside the reaction bottle. To ensure more accurate testing, a blank control method was adopted. Under the same conditions, a reaction bottle without hydrogen production bacteria was placed in the same equipment and instrument, and the temperature change of this reaction bottle was recorded. The temperature of the reaction solution in the adiabatic bottle was subtracted from the temperature of the control bottle, which is the change in system temperature caused by the actual metabolism of hydrogen production bacteria during the hydrogen production reaction of fermentation bacteria.

## The constant temperature system II

A regular reaction bottle with a reaction volume of 700 mL is used, and the reaction temperature is set in advance for the constant temperature chamber. When the temperature sensor senses a temperature change in the reaction bottle, the signal is transmitted to the computer. The computer then feeds back the temperature adjustment signal to the constant temperature chamber for temperature adjustment, keeping the entire temperature of the reaction bottle at the set temperature, ensuring that the hydrogen production system is not affected by accumulated heat.

The test conditions are set as follows: 25% of the inoculation amount, 3000 Lx of light, pH 7, and 10 g/L of glucose concentration for photo-fermentation hydrogen production system; 25% of the inoculation amount, pH 5.5, and 10 g/L of glucose concentration for dark fermentation hydrogen production system.

## Liquid rheological properties of baffle plate reactor and multiphase flow theory

The analytical methods were explained in the Supplementary Methods. The grid of the reacting region is shown in Supplementary Fig. 5.

## Continuous hydrogen production experiments

For dark and photo-fermentation hydrogen production system, 25 g/L reducing sugar hydrolyzed from corn straw was used as substrate. During the dark fermentation unit with three chambers (each chamber has a working volume of 1 m$^3$), the dark fermentation system was conducted under an initial pH of 5.5, HRT of 12 h, and 40 °C. The inoculation amount of dark fermentation hydrogen-producing bacteria was set to 20% (v/v) (cell dry weight 1.13 ± 0.15 g/L). The dark fermentation effluents flowing out of the dark fermentation unit first entered the treatment tank to remove excess $NH_4^+$ by zeolite adsorption. And then, the treated dark fermentation effluents (<3 mM $NH_4^+$) were pumped into the mixing chamber, the micronutrient solution and hydrogen production medium were added to the mixing chamber in proportion, and the pH was adjusted to 7.0. The mixture and inoculum (20% (v/v), cell dry weight 1.36 ± 0.15 g/L) were injected into the photo-fermentation unit by the pump. The fermentation conditions were set to a temperature of 30 ± 1 °C, light intensity of 3000 ± 200 lx, and HRT of 24 h, respectively.

The continuous bioreactor operation and monitoring is controlled via the automated control center designed by our team[38].

## Life cycle assessment method

The system boundary can be divided into three stages: 1. raw material pulverization; 2. enzymolysis pretreatment; 3. fermentation-based hydrogen production. According to ISO 14040 and ISO 14044, LCA for hydrogen production systems is made by using SimaPro 8.5.

## System boundary determination

This paper assesses hydrogen production systems, namely dark and photo-fermentation systems. The boundaries of the hydrogen production system are shown in Supplementary Fig. 10, including straw pulverization, enzymolysis pretreatment, and hydrogen production fermentation. The functional unit is 1 t $H_2$. The life cycle inventory is shown in Supplementary Table 1.

## Sensitivity analysis

Supplementary Fig. 12 shows the sensitivity analysis results.

As for the environmental impact in the life cycle of dark and photo fermentation, S1 has a more significant decrease in IR and ME than the decrease in S2. This makes it clear that cellulase contributes more to IR and ME and the effect decreases by 6.43% and 3.17%, respectively. Its effect on other items is weakened by 0.2–2.5%. The decline in solid-to-liquid ratio renders the weakening of life cycle environmental impact on FPMF to be most significant (9.28%), followed by FEP by 8.97%, IR, OFHH, FRS, and WC by 7%, OFTA, TE, FE, and HNCT by 6%, and other items by 2–5%.

By comparison, it can be found that the decrease in the solid-to-liquid ratio (S2) can better weaken the environmental impact in the life cycle of dark- and photo-fermentation hydrogen production system.

## Life cycle costing assessment method

Life cycle costing assessment covers all the costs arising from the whole hydrogen production stages, such as design, construction, production, operation, maintenance, and materials disposal. LCC assessment relies on the established system boundary, for change in the latter will affect the former. In this study, the LCC assessment adopts the same system boundary in LCA. The capital input for transforming biological straws into hydrogen production systems in this study is listed in Supplementary Fig. 13. The detailed methods were explained in the Supplementary Methods.

## Economic sensitivity analysis

Sensitivity analysis is a sort of uncertainty analysis approach that investigates the effect of a specific change in different uncertainty factors on the economic aspect of a project. According to the production cost, for dark and photo-fermentation systems, fixed asset investment, welfare, and reagent consumption have the major proportion of the total production cost. The sales income of the system is significantly affected by the sales prices of hydrogen and fermentation residual liquid. To observe the effect on the economic interests of the system, the range of selected parameters is set to be ±30%. The results variation of the financial NPV is as shown in Supplementary Fig. 14. For dark and photo-fermentation systems, an increase in initial investment, reagents consumption, welfare, and water/power charge would cause a decrease in the NPV. The rising hydrogen and fermentation effluent selling prices would also raise the NPV. When the hydrogen price goes from −30% to +30%, the financial NPV changes from 50,000 CNY to 1,118,800 CNY for dark and photo-fermentation systems.

## Reporting summary

Further information on research design is available in the Nature Portfolio Reporting Summary linked to this article.

## Data availability

Source data are provided with this paper. Source Data file has been deposited in Figshare under the accession code https://doi.org/10.6084/m9.figshare.24525253[62]. Source data are provided with this paper.

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

## Acknowledgements

This study was supported by the National Natural Science Foundation of China (No. 52076068 to Q.G.Z., 52206244 to Y.M.L., 52276183 to Z.P.Z., 52106240 to H.Z., and 52176184 to Y.Z.J.), Excellent Youth Science Foundation of Henan Province (No. 232300421063 to Z.P.Z.) and Scientific and Technological Innovation Talents plan in Colleges and Universities of Henan Province (No.22HASTIT024 to C.H. and 24 HASTIT023 to Z.P.Z.). We thank the Ph.D. (Ms.Tian Zhang, Mr. Yang Zhang, and Ms.Shengnan Zhu) and master's students of the biohydrogen production team for their help in data processing.

## Author contributions

Q.G.Z., Z.P.Z, and C.H. developed the study concept. Y.M.L., R.R., and J.Z.R. designed the methodology and built the model. Y.M.L., Z.P.Z., and H.Z. wrote the manuscript and Supplementary Materials. S.T., J.J.H., Y.Z.J., C.Y.L., and D.P.J. performed the analysis. Y.M., C.X.X., Y.W., Y.Y.J., X.T.Z., R.J.L., G.L., J.Z.Y., and N.T. revised the paper. All authors contributed to writing and reviewing the manuscript.

## Competing interests

The authors declare no competing interests.
