## [Peer Review File · Nature Communications]

REVIEWER COMMENTS

Reviewer #1 (Remarks to the Author):

The article examines the biological production of hydrogen through dark fermentation and dark photosynthesis, investigating scientific aspects such as spectrum coupling, thermal effect, and multiphase flow properties, as well as technical aspects like the environmental and economic impacts of this process.

The integration of dark fermentation coupled with photofermentation has attracted significant attention due to the advantage of enhancing H₂ yields and achieving higher substrate conversion efficiency. Various articles and reviews have discussed approaches to address challenges associated with integrating dark fermentation and photofermentation, such as optimization of operational parameters, designs, and configurations of bioreactors to enhance productivity. Among the key issues, discussions revolve around sterility problems in substrates when using microbial cultures, both pure and mixed, and the unstable hydrogen production resulting from metabolic shifts in bacteria. Another challenging problem in the context of large-scale biohydrogen production is its relatively high cost compared to fossil fuels, attributed to high raw material and manufacturing expenses.

In comparison with existing literature, the novelty of this study lies in investigating spectrum coupling, thermal effect, and multiphase flow properties in photofermentation. Within this thematic area, studies have explored these new perspectives on spectral band sources of light to optimize photofermentation. Filtering technology is used to couple solar energy entering the system with the spectral absorption characteristics of photosynthetic bacteria that produce hydrogen.

Regarding the writing:

The title of the work "Continuous biohydrogen production system: construction, operation, and sustainable evaluation" suggests that the complete system will be addressed, but here the focus was on optimizing the photofermentative fermentation system.

The introduction could be enhanced by more clearly highlighting the novelty and contribution of this work in studying a pilot-scale biological system coupled with variables such as spectral coupling, thermal effect, and multiphase flow properties. Also, equations 1, 2, and 3 are considered irrelevant, especially since they are not dealt with that type of substrate later on.

There is no clear discussion of why only variables related to the photosynthetic part are optimized, and nothing related to the dark fermentation is addressed.

In the theoretical basis section, it could be mentioned whether these acids are related to the bacteria used in the dark fermentation system and also photofermentation.

Reviewer #2 (Remarks to the Author):

This study delves into the construction, operation, and sustainable evaluation of a continuous biohydrogen production system, which holds great promise due to hydrogen's clean and renewable nature. The manuscript focuses on establishing pilot-scale biohydrogen production systems through dark and photo-fermentation processes and subsequently evaluates their sustainability. The analysis reveals that the evaluated system, producing 1 ton of H₂, consumes 171,530 MJ of energy, emits 9.37 t of CO₂ equivalent, and has a payback period of 6.86 years. The research also emphasizes the need for large-scale biological hydrogen production systems to drive commercialization, and it presents innovative approaches such as a multi-spectral coupling light supply system and optimized reactor design for improved efficiency. The findings offer insights into advancing biohydrogen production technology and its practical implementation, while also highlighting the significance of clean energy operation and lifecycle assessment. However, major revision is required before consideration, as shown below:

1- The title is not clear; here is a suggested title: Towards Sustainable Biohydrogen Production: Pilot-Scale Systems and Evaluation for Commercial Viability.

2- There are some typos that need to be fixed; please see my attachment.

3- The manuscript lacks a more detailed discussion on the potential challenges and limitations associated with the construction, operation, and sustainability of large-scale biohydrogen production systems, such as technical hurdles, economic feasibility, and scalability issues as reported in the work of “Critical challenges in biohydrogen production processes from the organic feedstocks. *Biomass Conv. Bioref.* 13, 8383–8401 (2023). <https://doi.org/10.1007/s13399-020-00965-x>”. This is a long with the estimation of biohydrogen production from agro-waste as in the work of “Assessment of bioenergy and syngas generation in India based on estimation of agricultural residues, *Energy Reports*, 9, 2023, 3771-3786”. This has to be highlighted and discussed in the revised manuscript.

4- How is your system is different than that system “Atmospheric carbon removal via industrial biochar systems: A techno-economic-environmental study”, which also used biomass waste, in terms of LCA and CO₂ eq. This system corresponds to a carbon removal capacity of 3.26 tCO₂e per hour and the removal of approximately 24,450 tCO₂e annually. The economic assessment revealed that the project is profitable; however, profitability is sensitive to pricing of the carbon removal service and biochar. A project internal rate of return (IRR) of 22.35% is achieved. The authors need to compare their approach with the literature in terms of LCA and profitability of using biomass waste.

5- the paper is that it could benefit from a more detailed discussion of the challenges of producing and storing hydrogen. While the paper does touch on these challenges, it does not provide a thorough analysis of the technical and economic difficulties associated with hydrogen production and storage. The authors could refer to the recent work of “Biofuel production, hydrogen production and water remediation by photocatalysis, biocatalysis and electrocatalysis. *Environ Chem Lett* (2023). <https://doi.org/10.1007/s10311-023-01581-7>”. This will add real value to the revised manuscript.

6- The bioconversion of biomass is not well written and should highlight the main constraints and findings from the most recent literature, as of “Biofuel production, hydrogen production and water remediation by photocatalysis, biocatalysis and electrocatalysis. Environ Chem Lett (2023). <https://doi.org/10.1007/s10311-023-01581-7>”.

7- The main limitations of the study, as well as useful future research directions, should be discussed in conclusion.

Reviewer #3 (Remarks to the Author):

The manuscript presents an assessment of biological hydrogen production through dark-and-photo fermentation, with a particular focus on the second. The authors conduct preliminary photo fermentation experiments at lab-scale, followed by operation of the full process at pilot-scale, and use the data for an energetic, environmental, and economic assessment.

Although the topic is of interest, I am afraid that the manuscript is not of sufficient quality to be published in Nature Communications.

GENERAL COMMENTS:

1. One of the main elements of novelty of the paper should be the pilot-scale demonstration of the process. Nonetheless, in my opinion, data related to just 20 days of operation in one single operating conditions (HRT, light intensity, temperature) are not enough to this purpose, especially because the operating conditions selected for the pilot plant operation are not justified or previously optimized.

2. The discussion and comparison with the literature is highly lacking: the reader has no indication whether the H₂ productivities obtained in this work are consistent with those obtained from other similar studies, even at smaller scale. Moreover, there is very little or no comparison with other sustainable H₂ production technologies. Considering that this is a very hot topic nowadays, a discussion to compare biological H₂ production with the other technologies that take part in the open debate (mostly green hydrogen via water electrolysis and biomass gasification) is fundamental. For example, an energy consumption of 171.5 MJ/kgH₂ is almost comparable to that required by green H₂ (nearly 200 MJ/kg H₂).

3. In the LCA, a GWP of 9.37 tCO₂eq/tH₂ is reported, but it is not clear if is of fossil or biogenic origin. Since the feedstock is biomass, probably the biogenic (direct) and fossil (indirect) emissions should be better emphasized. As presented, the process does not seem too competitive compared to conventional routes (SMR is characterized by nearly 10 tCO₂eq/tH₂). Again, the impacts of green hydrogen via water electrolysis and of biomass gasification are not compared and discussed.

4. It is not clear in the economic analysis what is the cost assumed for H₂ sale?

SPECIFIC COMMENTS

The rationale of the spectrum coupling and thermal effect analysis is not very clear.

1. Spectrum coupling: the absorption spectrum of the culture is assessed to determine which wavelengths are absorbed by the photosynthetic bacteria, and a number of peaks are identified. Then, growth experiments are conducted using specific wavelengths, which however do not match the peaks highlighted by the absorption spectrum. Then, different light sources are used (LEDs?), but their spectra is not shown to see how it matches the absorption spectrum. Also, an experiment with just white sunlight is missing, to understand whether using a filter or a coloured light source actually improves the performances. In any case, for all the tests the hydrogen yield is reported, however the production (m³/m³/d) is a more important information to discuss.

Minor comment: In Figure 2 b) and c) the text is too small to read, I suggest to improve the figure quality; in Figure 2 c), it would be better to use the same color for growth curve and istogram, to avoid confusion.

2. Thermal effect: This section is in general hard to follow, I would suggest to revise the text to make it more clear. In general, it is not clear to me what message we should get out of these results. Experimental observations are merely described, but without any comment or conclusion. For example, why do the author think in Figure 3 no heat release was observed at 33°C? Also, by looking at the results, my feeling is that, rather than thermal or not-thermal effect, the hydrogen production is influenced by the actual temperature, which seems to have an optimal value at nearly 30°C. Thus, it would probably be more relevant to simply show the effect of temperature on biomass growth and hydrogen production.

Reviewer #4 (Remarks to the Author):

The article titled "Continuous biohydrogen production system: 1 construction, operation, and sustainable evaluation" presents a comprehensive investigation into the potential of biohydrogen production

through “dark-and photo-fermentation” as a sustainable and efficient energy source. In the manuscript, a pilot-scale biohydrogen production system, which employs photofermentation coupled to dark fermentation, is described. The manuscript is divided into three parts: First, laboratory scale experiments were performed, in order to examine the effect of light quality and temperature regulation on the system. In the second part, experimental results from the 11 m³ pilot plant, which has been established in previous work, are shown. In the last part of the manuscript, the system is evaluated and life cycle assessments are performed. The energy consumption, CO₂ emissions, and payback period metrics provide a view of the environmental and economic implications of the proposed technology. The calculated energy consumption and emissions per unit of hydrogen produced serve as valuable insights for evaluating the overall efficiency and environmental impact of the process.

Overall, the article appears to contribute significantly to the field of biohydrogen production and its sustainability assessment. It tackles a relevant research gap, employs a multidisciplinary approach, and offers practical insights for both academia and industry. However, the manuscript contains some weaknesses, which have to be improved. See the comments to each line number of the manuscript below:

25: What is meant by cleanliness? Please explain or re-formulare the sentence

55: It should be referred to HyBECCS (hydrogen bioenergy with carbon capture and storage) as a possibility to contribute to “Carbon Dioxide Removal” which is a main advantage of biohydrogen production in terms of climate change mitigation

58: High hydrogen production and short processing times. Please ad the literature source for this statements and a reference?

104: Are sugar-containing waste and residue streams also possible for direct conversion? If not, could you give a perspective / an outlook for this?

117: Figure 1 is too small and the texts are not readable. Is it your own graphic or based on other references?

147: Figure is too small and texts not readable

129 and 133. Instead of “light waves”, “wavelengths” should be written.

In general, the description of the experimental conditions for this chapter “spectrum coupling” (p.5f) is not sufficient (also not in the Materials section) for the following reasons:

* The color scheme is confusing (and differs between Figure 2b and 2c). The color code should be consistent and reflecting the colors of the light employed (i.e. for Figure 2b: 400 nm light is violet, 470 nm blue-green, 540 nm green, 600 nm red, 700 nm dark red; for Figure 2c: for example the line in the graph for blue light should be blue and not red!)

* The filters used are mentioned in the Materials section. However, the company has to be mentioned there, too. A search in the internet showed that the filters were bought from insunoptics. On the homepage of this company, the transmission spectra of the filters (4 mm thickness) are shown, which explains some of the results, which the authors describe in their manuscript. In particular, for the

experiments shown in Figure 2b, it has to be taken into account, that the % transmittance differs for the filters used. For example, the transmittance of DTP700 is only approximately two-thirds of that from DTB600. Therefore, the low dry weight obtained with DTP700 could just be caused by the lower amount of photons, which reached the culture. In order to get meaningful results, the light intensities reaching the culture using the filters should be measured (with a quantum counter). Then, the results should be normalized for an equal amount of photons.

* Another weakness in the experiment shown in Figure 2b is, that the wavelengths between 800 and 1000 nm were not considered at all. In that wavelength region, the main absorption of the photosynthetic pigments occurs (i.e. the Qy band of the light-harvesting complex and reaction center bacteriochlorophylls of the photosynthetic units). The filter DTB720 has at least a weak absorbance in that region. So, an experiment using that filter should be done.

124: the absorption spectra of the photosynthetic pigments (i.e. bacteriochlorophylls and carotenoids) contain more than one absorption peak. So, it should be written: "...absorption peaks."

152 (Figure 2c): Which light sources did the authors use for the experiments shown? How did the spectrum look like? Were equal light intensities employed? Does "lamp" mean tungsten light?

Chapter "Thermal effect" (p. 7f): Here, it is very difficult (if not possible) to understand what the terms "I: Thermal effect" and "II: No thermal effect" mean. After reading the Materials section, it seems that that just means, that in condition I, the temperature was just monitored, and in condition II, the temperature was set to a constant value. This should be explained more clearly.

177-183 The lines should be re-formulated to be better understandable.

Page 19, Table 3: The costs should also be provide in US\$ to make the findings more comparable.

356: Correct error "Midpiont"

377: Maybe it could be pointed out what part of the CO₂ eq emissions are physical emissions of biogenic carbon dioxide (that could be captured for HyBECCS approaches) in order to calculate the maximum negative emission potential NEP

447/448: please check the proper usage of italics (italics only for strains, other words: normal)

456: scanned

457: Spectrophotometer from which company? Path-length of cuvette?

463: Filters from which company?

468: LEDs from which company? Spectra? Intensities?

* A short description about how the dry-weight determination and the hydrogen measurements were performed should be provided.

531: please check this sentence for correct English.

Please avoid repetitions throughout the document (e.g. 536 -550)

Response to reviewers

[Response to points raised by Reviewer#1]:

Point 1: The title of the work "Continuous biohydrogen production system: construction, operation, and sustainable evaluation" suggests that the complete system will be addressed, but here the focus was on optimizing the photo fermentation system.

Response: We thank you very much for your suggestion and the opportunity to improve our manuscript. We will be happy to edit the text further, based on helpful comments from the reviewers. In previous studies, it was found that dark fermentation hydrogen production system showed strong anti-interference ability, and the operation conditions have been optimized. According to your suggestion, we have added some content about optimizing the dark fermentation system in the revised Manuscript and Supplementary information, which were marked in red.

Point 2: The introduction could be enhanced by more clearly highlighting the novelty and contribution of this work in studying a pilot-scale biological system coupled with variables such as spectral coupling, thermal effect, and multiphase flow properties. Also, equations 1, 2, and 3 are considered irrelevant, especially since they are not dealt with that type of substrate later on.

Response: We thank you very much for your suggestions, we have revised the introduction to clearly highlight the novelty and contribution of this work, which were marked with red color. And equations 1, 2, and 3 have been removed.

In this article, a pilot scale dark and photo-fermentation biological hydrogen production device was constructed. Biohydrogen production is a complex biochemical reaction. The hydrogen production performance is closely relationship with light source and density, temperature fluctuations, and rheological properties significantly affecting the energy flow and mutual transmission of the system. Light provides the necessary photoelectrons for photosynthetic bacteria to generate adenosine triphosphate, but not all

wavelengths are suitable for the growth of photosynthetic bacteria and hydrogen production, the absorption of ineffective light can cause the “light saturation effect” resulting in lower hydrogen production performance. Biohydrogen production process is carried out with the participation of multiple enzymes in hydrogen production bacteria, which can be considered as a temperature sensitive biochemical reaction process due to the enzyme activity is significantly affected by temperature. For fermentation system, the temperature fluctuation is caused by radiation heat transfer of light, biological heat, and external environment. Therefore, the study of lighting supply strategy and temperature fluctuation is crucial in improving hydrogen production performance. In addition, the focus on the rheological characteristics of the fermentation system can achieve uniformity and stability of the system, improving the continuous hydrogen production capacity of the hydrogen production system.

Before the construction of the reactor, combining classical thermodynamic analysis methods with modern computational fluid dynamics methods, the distribution and transmission characteristics of light, heat, and mass of the fermentation system is analyzed, and based on these, the design of the dark and photo-fermentation biological hydrogen production is completed. As follow, first, a light supply system for the hydrogen production unit of photosynthetic fermentation was constructed, based on the spectral absorption characteristics of photosynthetic bacteria. Second, the temperature fluctuation of the fermentation system during the biological hydrogen production process were analyzed, providing a reference for the design of the temperature control system of the device.

Point 3: There is no clear discussion of why only variables related to the photosynthetic part are optimized, and nothing related to the dark fermentation is addressed.

Response: We thank you very much for your suggestion and the opportunity to improve our manuscript. In previous studies, it was found that dark fermentation hydrogen production system showed strong anti-interference ability, and the operation conditions

have been optimized. According to your suggestion, we have added some content about optimizing the dark fermentation system in the revised Manuscript and Supplementary information, which were marked in red.

Point 4: In the theoretical basis section, it could be mentioned whether these acids are related to the bacteria used in the dark fermentation system and also photofermentation.

Response: Thank you very much for your comments. Yes, the types of by-product acids are related to the bacteria, because different hydrogen production bacterial strains have different pathways for hydrogen production metabolism. The relevant content has been added in the theoretical basis section, which were marked in red.

such as the acetic acid route, the butyric acid route, and the ethyl alcohol route, **the specific metabolic pathway is related to the hydrogen producing bacteria used³⁵**. The small-molecular acids finally stay in the fermentation liquid. However, since dark fermentation bacteria cannot use small-molecular acids as a carbon source to produce hydrogen, the hydrogen yielded from dark fermentation remains low. Illuminated by sunlight, the photo fermentation bacteria can turn the small-molecular acids and sugar into hydrogen through inherent photosynthesis.

[Response to points raised by Reviewer#2]:

Point 1: The title is not clear; here is a suggested title: Towards Sustainable Biohydrogen Production: Pilot-Scale Systems and Evaluation for Commercial Viability.

Response: Thank you very much for your suggestion. The title "Continuous biohydrogen production system: construction, operation, and sustainable evaluation" was changed to " Towards Sustainable Biohydrogen Production: Pilot-Scale Systems and Evaluation for Commercial Viability "

Point 2: There are some typos that need to be fixed; please see my attachment.

Response: Thank you very much for your careful review. We have rechecked the article to modify the typos, all changes were marked in red.

Point 3: The manuscript lacks a more detailed discussion on the potential challenges and limitations associated with the construction, operation, and sustainability of large-scale biohydrogen production systems, such as technical hurdles, economic feasibility, and scalability issues as reported in the work of “Critical challenges in biohydrogen production processes from the organic feedstocks. *Biomass Conv. Bioref.* 13, 8383–8401 (2023). <https://doi.org/10.1007/s13399-020-00965-x>; This is a long with the estimation of biohydrogen production from agro-waste as in the work of “Assessment of bioenergy and syngas generation in India based on estimation of agricultural residues, *Energy Reports*, 9, 2023, 3771-3786”. This has to be highlighted and discussed in the revised manuscript.

Response: Thank you very much for your suggestion and the opportunity to improve our manuscript. We have added the detailed discussion on the potential challenges and limitations associated with the construction, operation, and sustainability of large-scale biohydrogen production systems in the revised manuscript, all changes were marked in red.

Dark and photo-fermentation technology, which shows higher substrate conversion efficiency and shorter hydrogen production time, has attracted more and more people. Some measures have been adopted to promote hydrogen yield from organic matters, such as screening of hydrogen producing bacteria, temperature regulation, optimization of reactors, addition of catalyst^{22,30}, etc., which were conducted in lab. But there is a lack of data on large-scale hydrogen production system. Pilot-scale biohydrogen production test is necessary prior to a commercial-scale application to analyze stable continuous operation feasibility and operation economic feasibility. Current research on these aforementioned technologies primarily relies on a kinetic model, and/or experimental data, to evaluate their industrial feasibility^{31–34}. Questions, however, remain on how to minimize the error for scale-up modelling, and to improve reactor flexibility and ease of operation. So, before further scale-up of these biohydrogen production

technologies can be achieved, more practical process parameters and feasibility evaluation studies need to be done. A better understanding of the scale-up modelling would significantly improve the process design scale-up decision-making.

In this article, a pilot scale dark and photo-fermentation biological hydrogen production device was constructed. Biohydrogen production is a complex biochemical reaction. The hydrogen production performance is closely relationship with light source and density, temperature fluctuations, and rheological properties significantly affecting the energy flow and mutual transmission of the system. Light provides the necessary photoelectrons for photosynthetic bacteria to generate adenosine triphosphate, but not all wavelengths are suitable for the growth of photosynthetic bacteria and hydrogen production, the absorption of ineffective light can cause the “light saturation effect” resulting in lower hydrogen production performance. Biohydrogen production process is carried out with the participation of multiple enzymes in hydrogen production bacteria, which can be considered as a temperature sensitive biochemical reaction process due to the enzyme activity is significantly affected by temperature. For fermentation system, the temperature fluctuation is caused by radiation heat transfer of light, biological heat, and external environment. Therefore, the study of lighting supply strategy and temperature fluctuation is crucial in improving hydrogen production performance. In addition, the focus on the rheological characteristics of the fermentation system can achieve uniformity and stability of the system, improving the continuous hydrogen production capacity of the hydrogen production system.

Before the construction of the reactor, combining classical thermodynamic analysis methods with modern computational fluid dynamics methods, the distribution and transmission characteristics of light, heat, and mass of the fermentation system is analyzed, and based on these, the design of the dark and photo-fermentation biological hydrogen production is completed. As follow, first, a light supply system for the hydrogen production unit of photosynthetic fermentation was

constructed, based on the spectral absorption characteristics of photosynthetic bacteria.

For a continuous hydrogen production system, the stable and continuous generation of hydrogen is an important goal, which may be achieved under high organic loading rate, leading to the loss of hydrogen producing bacteria and a certain amount of organic matter remains in the tail liquid after fermentation. Therefore, some technologies should be adopted to treat the hydrogen production tail liquid to achieve efficient utilization of substrates, such as preparation of liquid organic fertilizer, and methane production from hydrogen production tail liquid.

Point 4: How is your system is different than that system “Atmospheric carbon removal via industrial biochar systems: A techno-economic-environmental study”, which also used biomass waste, in terms of LCA and CO₂ eq. This system corresponds to a carbon removal capacity of 3.26 tCO₂e per hour and the removal of approximately 24,450 tCO₂e annually. The economic assessment revealed that the project is profitable; however, profitability is sensitive to pricing of the carbon removal service and biochar. A project internal rate of return (IRR) of 22.35% is achieved. The authors need to compare their approach with the literature in terms of LCA and profitability of using biomass waste.

Response: We thank you very much for your kind suggestions. We have added the comparison content, all changes were marked in red.

The GW for hydrogen production system, the GW was mainly originated from hydrogen production process, was over 60% of the total amount. The reason may be due to the addition of chemical reagents during the fermentation stage. This provides a research direction for us to reduce GW from biological hydrogen production processes. Supplementary Table.2 shows the GW from different hydrogen production technology. The GW of the hydrogen production via biomass gasification and coal gasification is approximately 10.56 kg CO₂ eq/kg H₂⁴⁵ and 18 kg CO₂ eq/kg⁴⁶, respectively. The GW of green hydrogen production via water electrolysis using wind power or solar power

shows lower value varying from 9.4 to 0.3 kg CO₂ eq/kg^{47,48}. But renewable power generation is limited by land area available for photovoltaic panels and/or wind turbines, when the grid electricity is used to compensate for insufficient wind or solar electricity, the GW could reach to 25.93 CO₂ eq/kg H₂⁴⁹. Comparing to green hydrogen production via water electrolysis, biological fermentation hydrogen production system exhibits advantages in resource recycling and simultaneously achieving waste treatment and clean energy production. With the application of carbon capture and storage technology, the preparation process of materials gradually becomes cleaner, which will help reduce the GW of the biological fermentation hydrogen production system.

Through literature comparison, it was also found that different biomass conversion technologies exhibit different emission reduction capabilities based on LCA⁵⁰⁻⁵², caused by the different in assessment methods, functional units and system boundaries⁵³, but the final results all indicate development and utilization of biomass helps to net-zero emissions.

Point 5: The paper is that it could benefit from a more detailed discussion of the challenges of producing and storing hydrogen. While the paper does touch on these challenges, it does not provide a thorough analysis of the technical and economic difficulties associated with hydrogen production and storage. The authors could refer to the recent work of “Biofuel production, hydrogen production and water remediation by photocatalysis, biocatalysis and electrocatalysis. Environ Chem Lett (2023). <https://doi.org/10.1007/s10311-023-01581-7>”; This will add real value to the revised manuscript.

Response: Thank you very much for your suggestions. This article focuses on the design and operation of continuous reactors, and the purification of biogenic gases is not within the boundaries of life cycle assessment. That is a good suggestion, providing us with a good direction. We have added the discussion about the challenges of producing and storing hydrogen.

Point 6: The bioconversion of biomass is not well written and should highlight the main constraints and findings from the most recent literature, as of “Biofuel production, hydrogen production and water remediation by photocatalysis, biocatalysis and electrocatalysis. Environ Chem Lett (2023). <https://doi.org/10.1007/s10311-023-01581-7>”;

Response: Thank you very much for your suggestions. We have added the content to highlight the main constraints and findings from the most recent literature about biohydrogen production. All changes were marked in red.

Most of these conventional approaches consume non-renewable resources as the raw materials, which not only **increases** the hydrogen production costs, but also **exacerbates** environmental **issues due to** emissions of harmful pollutants. **At present, hydrogen is given different colour shades(black, grey, brown, grey and green) according to the energy source used and effects on environment during hydrogen production technology¹⁰. Compared with conventional hydrogen production, biohydrogen production method utilizes microorganisms, resulting in fermentation or photosynthesis, to release free hydrogen gas¹¹⁻¹³, which is a more promising approach for preparing green hydrogen due to independent of finite resources and contribute to “Carbon Dioxide Removal”¹⁴⁻¹⁷. Biological approaches include biophotolysis (direct and indirect), electrochemical system(microbial electrolysis cell), biological fermentation (dark fermentation, photo fermentation, and dark and photo- fermentation)¹⁸⁻²². Among them, biological fermentation with the advantages of wider range of available raw materials (crop straw, livestock manure, kitchen waste) and potentially lower production costs has become a research focus.**

Dark fermentation hydrogen production is the process in which bacteria convert organic matter into hydrogen and volatile fatty acids **in the absence of light**^{23,24}. This process is characterized by **high hydrogen production rate** and short processing **times**²³. However, a large amount of volatile fatty acids remain in the fermentation effluent, resulting in low substrate conversion efficiency. In comparison, during photo fermentation hydrogen production, photosynthetic bacteria convert monosaccharides and **some volatile fatty acids into hydrogen**^{25,26}, under conditions exposed to light. This process features high substrate conversion and mild reaction temperatures.

Combining the two mechanisms described above, dark fermentation can be used, first,

for hydrogen production, and the remaining substrate can be fed to photosynthetic bacteria, thus creating a two-step “dark and photo-fermentation”, to improve the overall substrate conversion efficiency^{26–29} Dark and photo-fermentation technology, which shows higher substrate conversion efficiency and shorter hydrogen production time, has attracted more and more people. Some measures have been adopted to promote hydrogen yield from organic matters, such as screening of hydrogen producing bacteria, temperature regulation, optimization of reactors, addition of catalyst^{22,30}, etc., which were conducted in lab. But there is a lack of data on large-scale hydrogen production system. Pilot-scale biohydrogen production test is necessary prior to a commercial-scale application to analyze stable continuous operation feasibility and operation economic feasibility. Current research on these aforementioned technologies primarily relies on a kinetic model, and/or experimental data, to evaluate their industrial feasibility^{31–34}. Questions, however, remain on how to minimize the error for scale-up modelling, and to improve reactor flexibility and ease of operation.

Point 7: The main limitations of the study, as well as useful future research directions, should be discussed in conclusion.

Response: Thank you very much for your comments. According to your suggestion, we have added the relevant information in the manuscript, which were marked in red.

The construction and operation of a large-scale biological hydrogen production system is a necessary step to achieve the commercialization of biological hydrogen production. In this article, a multi-spectral coupling light supply system, with solar energy as the energy source, was constructed, based on the light absorption characteristics of the hydrogen producing microorganisms. The bio-hydrogen production reactor was optimized according to the thermal effect and multiphase flow characteristics of the fermentation system. Based on the above theoretical foundations, we have constructed a pilot scale biological hydrogen production reactor that relies entirely on clean energy operation. Evaluation and analysis results show when the system produces 1 ton of H₂, it will consume 171,530 MJ of energy and emit greenhouse gas (GHG) 9.37 t CO₂ eq.

In the sensitivity analysis of the lifecycle assessment of the system, it was found that increasing the solid-liquid ratio of the fermentation system can reduce greenhouse gas emissions during the hydrogen production process. The theoretical research and fermentation equipment design and operation in the paper may provide theoretical and equipment support for the development of biomass green hydrogen technology, and provide systematic solutions for carbon sequestration and emission reduction in the industrial and agricultural fields, supporting the development of carbon neutrality and circular economy.

[Response to points raised by Reviewer#3]:

Point 1: One of the main elements of novelty of the paper should be the pilot-scale demonstration of the process. Nonetheless, in my opinion, data related to just 20 days of operation in one single operating conditions (HRT, light intensity, temperature) are not enough to this purpose, especially because the operating conditions selected for the pilot plant operation are not justified or previously optimized.

Response: Thank you very much for your comments. The influencing factors of continuous operation experiments have been optimized in previous studies which have been cited, here the fermentation system was operated under the optimal fermentation process. The hydrogen production device has been running steadily for two years continuously. For the convenience of observing the data, we have only presented data from some time periods. Some relevant discussions were added for the convenience of reading, all changes were marked in red.

Enzymatic hydrolysate of corn straw(reducing sugar) is used, at a concentration of 25 g/L, as the substrate for hydrogen production via the dark and photo-fermentation hydrogen production device. First, reactor was operated in batch mode for 30 d to enrich the functional strains, and then the system was switched to continuous model. After two weeks of operation, the system tends to stabilize. During the two-year operation period, the operating process was optimized^{36,38,39}. Once we have collected data for 20 days of stable operation, as shown in Fig.7b, the average hydrogen production rate of the dark fermentation unit is 15.04 m³/m³-d, and the average hydrogen production rate of the photosynthetic fermentation is 8.26 m³/m³-d. Through calculation, it can be found that the device can produce 10 kg of hydrogen per day. The average hydrogen production rate obtained in the paper was higher than that of semi-pilot scale up-flow anaerobic sludge blanket reactor⁴⁰ and packed bed biofilm reactor⁴¹. For the pilot-scale hydrogen production system in the paper, each chamber is a baffle reactor and can independently ferment for hydrogen production, which shows good flexibility and scalability to meet the construction of fermentation systems of different scales. For a continuous hydrogen production system, the stable and continuous generation of hydrogen is an important goal, which may be achieved under high organic loading rate, leading to the loss of residual hydrogen producing bacteria and a certain amount of organic matter in the tail liquid after fermentation. Therefore, some technologies should be adopted to treat the hydrogen production tail liquid to achieve efficient utilization of substrates, such as preparation of liquid organic fertilizer, and methane production from hydrogen production tail liquid.

Point 2: The discussion and comparison with the literature is highly lacking: the reader has no indication whether the H₂ productivities obtained in this work are consistent with those obtained from other similar studies, even at smaller scale. Moreover, there is very little or no comparison with other sustainable H₂ production technologies. Considering that this is a very hot topic nowadays, a discussion to compare biological H₂ production with the other technologies that take part in the open debate (mostly green hydrogen via water electrolysis and biomass gasification) is fundamental. For example, an energy consumption of 171.5 MJ/kgH₂ is almost comparable to that

required by green H₂ (nearly 200 MJ/kg H₂).

Response: Thank you very much for your suggestions and opportunities to improve manuscripts. We have added the discussion and comparison marked with red color.

During the fermentation stage, the energy consumption mainly comes from feeding devices, lighting devices, mixing devices, and online monitoring systems. Results show that the dark and photo-fermentation system consumes 171,530 MJ to generate 1 t hydrogen (171.53 MJ/kg H₂), this is supplied by solar energy, which is close to the energy consumption of green hydrogen production by electrolysis water using solar and wind energy (183.58 MJ/kg H₂⁴², 150.10 MJ/kg H₂⁴³, 168.82 MJ/kg H₂⁴³). But the energy consumption is lower than that of hydrogen production via using deep in-situ gasification based coal-to-hydrogen (349.55 MJ/kg H₂)⁴⁴. During the operation, energy consumption mainly occurs in the enzymolysis stage at 53.04%, the pulverization stage at 9.76%, and for the fermentation stage at 37.2%. In the future, the research on high activity cellulase and high solid-phase enzyme hydrolysis technology should be followed to reduce to reduce the energy consumption of enzyme hydrolysis processes.

Energy consumption is one resulting aspect on the environment from the hydrogen production system. To analyze the environmental impact of such systems in detail, it is necessary to perform a life cycle environmental assessment. The following sections will elaborate this assessment for hydrogen production system.

Point 3: In the LCA, a GWP of 9.37 tCO₂eq/tH₂ is reported, but it is not clear if it is of fossil or biogenic origin. Since the feedstock is biomass, probably the biogenic (direct) and fossil (indirect) emissions should be better emphasized. As presented, the process does not seem too competitive compared to conventional routes (SMR is characterized by nearly 10 tCO₂eq/tH₂). Again, the impacts of green hydrogen via water electrolysis and of biomass gasification are not compared and discussed.

Response: Thank you very much for your comments. The GWP comes from fossil (indirect), due to fossil was consumed to production of materials required for reactor construction. We have added comparative analysis with other technologies in the Manuscript and Supplementary information, which were marked in red

The GW for hydrogen production system, the GW was mainly originated from hydrogen production process, was over 60% of the total amount. The reason may be due to the addition of chemical reagents during the fermentation stage. This provides a research direction for us to reduce GW from biological hydrogen production processes. Supplementary Table.2 shows the GW from different hydrogen production technology. The GW of the hydrogen production via biomass gasification and coal gasification is approximately 10.56 kg CO₂ eq/kg H₂⁴⁵ and 18 kg CO₂ eq/kg⁴⁶, respectively. The GW of green hydrogen production via water electrolysis using wind power or solar power shows lower value varying from 9.4 to 0.3 kg CO₂ eq/kg^{47,48}. But renewable power generation is limited by land area available for photovoltaic panels and/or wind turbines, when the grid electricity is used to compensate for insufficient wind or solar electricity, the GW could reach to 25.93 CO₂ eq/kg H₂⁴⁹. Comparing to green hydrogen production via water electrolysis, biological fermentation hydrogen production system exhibits advantages in resource recycling and simultaneously achieving waste treatment and clean energy production. With the application of carbon capture and storage technology, the preparation process of materials gradually becomes cleaner, which will help reduce the GW of the biological fermentation hydrogen production system.

Through literature comparison, it was also found that different biomass conversion technologies exhibit different emission reduction capabilities based on LCA⁵⁰⁻⁵², caused by the different in assessment methods, functional units and system boundaries⁵³, but the final results all indicate development and utilization of biomass helps to net-zero emissions.

Supplementary Table.2 GW from different hydrogen production technology

Hydrogen production method	Global warming t CO ₂ eq/t H ₂	References
CLC-SR	10.76	4
Underground coal gasification	18	5
Deep IGtH	30.47	6
Lurgi SGtH	36.41	6
Biomass gasification	10.56	7

Coal Gasification	11.30	8
ESR-bioeth	9.2	9
SMR-F.M	11.2	9
SMR	15	10
DF-MEC	17	11
DPFHP	9.37	This study

CLC-SR: Chemical looping combustion thermally coupled steam reforming

Lurgi SGCtH: Lurgi surface gasification based coal-to-hydrogen

Deep IGCtH: deep in-situ gasification based coal-to-hydrogen

ESR-bioeth : Bioethanol into hydrogen steam reforming

SMR-F.M (Fossil-methane steam reforming)

DPFHP: Dark and photo-fermentation hydrogen production

DF-MEC :Dark fermentation-microbial electrolysis cell

Point 4: It is not clear in the economic analysis what is the cost assumed for H2 sale?

Response: Thank you very much for your suggestions. We have modified the section, marked with red color.

When the (net present value) $NPV > 0$, it means the plan is economically feasible. T_p is the investment payback period, IRR is the internal return of rate. When IRR is higher than the benchmark discount rate (8%), the project has a favorable economic effect.³¹

According to market price at project site, the sale price of hydrogen is set 56 CHY/kg(7.65\$/kg). Based on the data of Table 3, the financial net present value (NPV) of dark and photo- fermentation system is 584,400 CNY(79,843\$/kg), the investment payback periods are estimated to be 6.86 years and IRR is 16.84% for dark- and photo-fermentation system(Supplementary Table 4), the payback period is lower than 10.28 years for hydrogen production with water electrolysis (IRR 10.28%)⁵⁵.

Point 5: Spectrum coupling: the absorption spectrum of the culture is assessed to determine which wavelengths are absorbed by the photosynthetic bacteria, and a number of peaks are identified. Then, growth experiments are conducted using specific wavelengths, which however do not match the peaks highlighted by the absorption spectrum. Then, different light sources are used (LEDs?), but their spectra is not shown to see how it matches the absorption spectrum. Also, an experiment with just white

sunlight is missing, to understand whether using a filter or a coloured light source actually improves the performances. In any case, for all the tests the hydrogen yield is reported, however the production ($\text{m}^3/\text{m}^3/\text{d}$) is a more important information to discuss. Minor comment: In Figure 2 b) and c) the text is too small to read, I suggest to improve the figure quality; in Figure 2 c), it would be better to use the same color for growth curve and istogram, to avoid confusion.

Response: Thank you very much for your comments. Yes, growth experiments are conducted using specific wavelengths, which do not match the peaks highlighted by the absorption spectrum. Although some wavelengths of light can be absorbed by photosynthetic bacteria, not all absorbed light can promote bacterial growth and hydrogen production. During experiments, 400 nm and 720 nm show negative impact on hydrogen yield, so, light below 400 nm or above 720 nm is not further used. We chose the wavelengths near the absorption peak in visible light (390~760 nm), which are easily accessible.

During different light sources(LED) experiments, the absorption wavelength is basically within the wavelength range of the selected light source: yellow light (597~577 nm), green light (577~492 nm), blue light (492~455 nm), red light (780~622), white light (450~465). An experiment with white light was conducted, the light source was written as lamp, we have revised the lamp to white light.

In spectrum coupling experiments, all tests were carried out in batch mode to achieve optimal conditions in term of hydrogen yield, therefore, the hydrogen production rate($\text{m}^3/\text{m}^3/\text{d}$) was not analyzed.

The figures quality have been improved, colors for growth curve also have been changed.

Fig.2 Characteristics of the absorption of photosynthetic bacteria(a), photosynthetic bacteria growth and hydrogen production under different wavelengths (b) and photosynthetic bacteria growth and hydrogen production under different light sources (c)

Point 6: Thermal effect: This section is in general hard to follow, I would suggest to revise the text to make it more clear. In general, it is not clear to me what message we should get out of these results. Experimental observations are merely described, but without any comment or conclusion. For example, why do the author think in Figure 3 no heat release was observed at 33°C? Also, by looking at the results, my feeling is that, rather than thermal or not-thermal effect, the hydrogen production is influenced by the actual temperature, which seems to have an optimal value at nearly 30°C. Thus, it would probably be more relevant to simply show the effect of temperature on biomass growth and hydrogen production.

Response: Thank you very much for your suggestions, I'm very sorry for not explaining clearly, which makes it inconvenient for you to read. The thermal effect analysis is to provide reference for the design of the temperature control system of the reactor. We have added comments and revised the section for better understanding, all changes were marked in red.

Thermal effect: During the hydrogen production by photosynthetic bacteria, light irradiates the reaction system through the reactor. Part of the physical factors accepted by photosynthetic bacteria is converted into thermal energy and released directly, and part of the physical factors is absorbed by the biological system to increase the cell's metabolism with the released thermal energy. For dark fermentation, only the latter occurs. The biological effects caused by the above impact are the thermal effect of

hydrogen production of fermentation bacteria. The generated heat leads to increased temperature, which affects the strength of the enzyme activity, and then affects the hydrogen production efficiency. The experiments are designed to analyze the thermal effect of different initial temperatures and light intensities on hydrogen production, to provide reference for the design of temperature control systems in pilot-scale reactors. For thermal effect experiments, the temperature-varying system experiment is carried out in vacuum reaction bottle, the constant temperature system experiment is performed in regular glass bottles maintained at a constant temperature realized by the temperature control system. The reaction system involves several factors, such as the biochemical reaction, light source, and temperature control. (Supplementary Fig.3 shows heat energy transmission of the system. Supplementary Fig.4 shows illustration of experiment equipment of the hydrogen production system on thermal effect.)

The temperature fluctuation of temperature varying system at different initial temperatures for dark fermentation system and photo fermentation system is shown in Figs.3a and 3d. During the first 12 hours of fermentation, the system temperature at different initial temperatures increase to a large extent. The system temperature rises slowly from 12 to 20 hours, and remains basically stable after 20 hours. The variation rate of the temperature does not show regularity with the change of the initial temperature for fermentation system, but variation rate of the temperature for photo fermentation is greater than that of dark fermentation, which may be caused by light radiation. The maximum temperature increment is 3.43°C, detected in photo fermentation system with initial temperature of 27°C. The magnitude of the temperature change is 36°C, 33°C, 39°C and 30°C for dark fermentation and is 27°C, 30°C, 24°C, and 33°C for photo fermentation. The heat production rate at different initial temperatures is shown in Figs. 3b and 3e. The heat production rate increases rapidly from 2 to 6 hours for dark fermentation system, and then decreases until the system reaches heat balance, but for photo fermentation system, the heat production rate increases rapidly from 2 to 8 hours, and then decreases. The reason can be explained by the fact that dark fermentation bacteria can quickly adapt to fermentation environment

to grow and produce hydrogen. The maximum heat generation rate(1.14kJ/L·h) is detected in photo fermentation system with the initial temperature of 27°C. Figs. 3c and 3f shows the hydrogen production at different initial temperatures. At 30 °C and 33°C in dark fermentation system, 24 and 27 °C in photo fermentation system, the hydrogen production is affected by temperature fluctuations, and is greater than that of constant temperature system at the same initial temperature. The opposite situation occurs when the initial temperature is 36 and 39°C in dark fermentation system, 30 and 33°C in photo fermentation system. The reason may be due to the activity of the enzyme is negatively affected by the temperature increasing. The maximum hydrogen production occurs at 27°C while the minimum occurs at 24°C for photo fermentation system.

[Response to points raised by Reviewer#4]:

Point 1: What is meant by cleanliness? Please explain or re-formulare the sentence

Response: Thank you very much for your suggestions. We have modified the sentence marked with red color.

Featuring high caloric value, **clean-burning**, and renewability, hydrogen is a **fuel believed** to be able to change energy structure worldwide.

Point 2: It should be referred to HyBECCS (hydrogen bioenergy with carbon capture and storage) as a possibility to contribute to “Carbon Dioxide Removal” which is a main advantage of biohydrogen production in terms of climate change mitigation.

Response: Thank you very much for your comments. We have modified the sentence marked with red color.

Most of these conventional approaches consume non-renewable resources as the raw materials, which not only **increases** the hydrogen production costs, but also **exacerbates** environmental **issues due to** emissions of harmful pollutants. **At present, hydrogen is given different colour shades(black, grey, brown, grey and green) according to the energy source used and effects on environment during hydrogen production technology¹⁰. Compared with conventional hydrogen production, biohydrogen production method utilizes microorganisms, resulting in fermentation or photosynthesis, to release free hydrogen gas¹¹⁻¹³, which is a more promising approach for preparing green hydrogen due to independent of finite resources and contribute to “Carbon Dioxide Removal” ¹⁴⁻¹⁷. Biological approaches include biophotolysis (direct and indirect), electrochemical system(microbial electrolysis cell), biological fermentation (dark fermentation, photo fermentation, and dark and photo- fermentation) ¹⁸⁻²². Among them, biological fermentation with the advantages of wider range of available raw materials (crop straw, livestock manure, kitchen waste) and potentially lower production costs has become a research focus.**

Point 3: High hydrogen production and short processing times. Please add the literature source for this statements and a reference?

Response: Thank you very much for your suggestions. We have added a reference, changes marked in red color.

Dark fermentation hydrogen production is the process in which bacteria convert organic matter into hydrogen and volatile fatty acids **under dark conditions** ^{23,24}. This process is characterized by **high hydrogen production rate** and short processing **times**²³. However, a large amount of volatile fatty acids remain in the fermentation effluent, resulting in low substrate conversion efficiency. In comparison, during photo fermentation hydrogen production, photosynthetic bacteria convert monosaccharides and **some volatile fatty acids into hydrogen** ^{25,26}, under conditions exposed to light. This process features high substrate conversion and mild reaction temperatures.

Point 4: Are sugar-containing waste and residue streams also possible for direct conversion? If not, could you give a perspective / an outlook for this?

Response: Thank you very much for your comments. Yes, sugar-containing waste and residue streams can be directly utilized for hydrogen production, the hydrogen production performance of molasses wastewater and apple waste was studied in previous studies(Bio-hydrogen production from apple waste by photosynthetic bacteria HAU-M1 <http://dx.doi.org/10.1016/j.ijhydene.2016.06.101>), higher hydrogen yield was obtained than that of from corn straw.

Point 5: Figure 1 is too small and the texts are not readable. Is it your own graphic or based on other references?

Response: Thank you very much for your suggestions. The Figure 1 has been modified. Yes. the figure is our own graphic.

Fig.1 Biochemical process of hydrogen production

Point 6: Figure is too small and texts not readable.

Response: Thank you very much for your comments. The all figures have been modified.

Point 7: 129 and 133. Instead of “light waves”, “wavelengths” should be written.

In general, the description of the experimental conditions for this chapter “spectrum coupling” (p.5f) is not sufficient (also not in the Materials section) for the following reasons:

* The color scheme is confusing (and differs between Figure 2b and 2c). The color code should be consistent and reflecting the colors of the light employed (i.e. for Figure 2b: 400 nm light is violet, 470 nm blue-green, 540 nm green, 600 nm red, 700 nm dark red; for Figure 2c: for example the line in the graph for blue light should be blue and not red!)

Response: Thank you very much for your suggestions. We have changed “light waves” to “wavelengths”.

Thank you for your suggestion color scheme, we have changed color code, which is can reflect the colors of the light employed. But for red and green, the similar colors are adopted to ensure some people those colour-vision deficiency can read.

Then, the filters are used to screen the **wavelengths** suitable for the growth of

photosynthetic bacteria and hydrogen production to eliminate the “light saturation effect” caused by absorption of excessive light energy and obtain the coupling between photosynthetic bacteria and solar energy. The growth characteristics and hydrogen production effect of photosynthetic bacteria under different wavelengths are shown in Fig. 2b. The growth effect of photosynthetic bacteria is best under the 400 nm spectrum; the maximum cell dry weight can reach 3.55 g/L with the minimum hydrogen production.

Fig.2 Characteristics of the absorption of photosynthetic bacteria(a), photosynthetic bacteria growth and hydrogen production under different wavelengths (b) and photosynthetic bacteria growth and hydrogen production under different light sources(c)

Point 8: The filters used are mentioned in the Materials section. However, the company has to be mentioned there, too. A search in the internet showed that the filters were bought from insunoptics. On the homepage of this company, the transmission spectra of the filters (4 mm thickness) are shown, which explains some of the results, which the

authors describe in their manuscript. In particular, for the experiments shown in Figure 2b, it has to be taken into account, that the % transmittance differs for the filters used. For example, the transmittance of DTP700 is only approximately two-thirds of that from DTB600. Therefore, the low dry weight obtained with DTP700 could just be caused by the lower amount of photons, which reached the culture. In order to get meaningful results, the light intensities reaching the culture using the filters should be measured (with a quantum counter). Then, the results should be normalized for an equal amount of photons.

Response: Thank you very much for your comments. The filters were purchased from Nantong Yinxing Optics Co., Ltd, China, the relevant information has been added.

Yes, different types of filters show different transmittance, during experiments the illumination intensity through filters is set to the same value(3000Lx) which is also the optimal illumination intensity for hydrogen production bacteria growth. The operating conditions of the experiment were described in the Methods.

Experiment on the Influence of Spectra and Light Sources on Hydrogen

Production: Using the light from the xenon lamp that is closest to the solar spectrum as the light source, a filter device is installed in front of the light collector to obtain spectra of different wavelengths. The filters used in this article are 100 nm bandpass filters DTB400, DTB470, DTB540, DTB600, DTB700, and DTB720(Nantong Yinxing Optics Co., Ltd, China) to process the collected xenon lamp light, in order to ensure the uniformity of photosynthetic bacteria receiving light and avoid losses caused by light scattering, the xenon light is collected by a collector and transmitted to the glass bottle by optical filters(the reaction device is shown in Supplementary Figure 2a).

When conducting experiments with different light sources, the light sources used include yellow light source, blue light source, green light source, red light source and incandescent light source. To control a certain temperature, the reaction bottle is placed in a constant temperature box, and each group of experiments is conducted three times (the reaction device is shown in Supplementary Figure 2b).

The test conditions are set as follows: 25% of the inoculation amount, 30 °C of temperature, 3000 Lx of light, pH 7, and 10 g/L of glucose concentration.

Point 9: Another weakness in the experiment shown in Figure 2b is, that the wavelengths between 800 and 1000 nm were not considered at all. In that wavelength region, the main absorption of the photosynthetic pigments occurs (i.e. the Qy band of the light-harvesting complex and reaction center bacteriochlorophylls of the photosynthetic units). The filter DTB720 has at least a weak absorbance in that region. So, an experiment using that filter should be done.

Response: Thank you very much for your comments. We will be happy to edit the text further, based on helpful comments from the reviewers. An experiment with filter DTB720 has been done, the results are showed in **Fig.2b**.

Fig.2 photosynthetic bacteria growth and hydrogen production under different wavelengths (b)

Point 10: 124: the absorption spectra of the photosynthetic pigments (i.e. bacteriochlorophylls and carotenoids) contain more than one absorption peak. So, it should be written: "...absorption peaks."

Response: Thank you very much for your comments. We have revised "absorption peak" to "absorption peaks".

Point 11: 152 (Figure 2c): Which light sources did the authors use for the experiments shown? How did the spectrum look like? Were equal light intensities employed? Does “lamp” mean tungsten light?

Response: Thank you. The LED light sources was used, the light intensity is set to 3000Lx for all experiments. No, the “lamp” means white light, we have revised lamp to white.

Point 12: Chapter “Thermal effect” (p. 7f): Here, it is very difficult (if not possible) to understand what the terms “I: Thermal effect” and “II: No thermal effect” mean. After reading the Materials section, it seems that that just means, that in condition I, the temperature was just monitored, and in condition II, the temperature was set to a constant value. This should be explained more clearly.

Response: Thank you very much for your comments. Your understanding is correct.

We have revised the section to explain more clearly. All changes ware marked in red/

Thermal effect: The experiments are designed to analyze the thermal effect of different initial temperatures and light intensities on hydrogen production, proving to provide reference for the design of temperature control systems in pilot-scale reactors. For thermal effect experiments, the temperature-varying system experiment is carried out in vacuum reaction bottle, the constant temperature system experiment is performed in regular glass bottles maintained at a constant temperature realized by the temperature control system. The reaction system involves several factors, such as the biochemical reaction, light source, and temperature control. (Supplementary Fig.3 shows heat energy transmission of the system. Supplementary Fig.4 shows illustration of experiment equipment of the hydrogen production system on thermal effect.)

Point 13: 177-183 The lines should be re-formulated to be better understandable.

Response: Thank you very much for your comments. lines 177-183 have been re-formulated, changes are marked in red.

Figs.3a and 3d. During the first 12 hours of fermentation, the system temperature at

different initial temperatures increase to a large extent. The system temperature rises slowly from 12 to 20 hours, and remains basically stable after 20 hours. The variation rate of the temperature does not show regularity with the change of the initial temperature for **fermentation system**, but variation rate of the temperature for photo fermentation is greater than that of dark fermentation, which may be caused by light radiation. **The maximum temperature increment is 3.43°C, detected in photo fermentation system with initial temperature of 27°C.** The magnitude of the temperature change is 36°C, 33°C, 39°C and 30°C for dark fermentation and is 27°C, 30°C, 24°C, and 33°C for photo fermentation. The heat production rate at different initial temperatures is shown in Figs. 3b and 3e. **The heat production rate increases rapidly from 2 to 6 hours for dark fermentation system, and then decreases until the system reaches heat balance, but for photo fermentation system, the heat production rate increases rapidly from 2 to 8 hours, and then decreases.** The reason can be explained by the fact that dark fermentation bacteria can quickly adapt to fermentation environment to grow and produce hydrogen. **The maximum heat generation rate(1.14kJ/L·h) is detected in photo fermentation system with the initial temperature of 27°C.** Figs. 3c and 3f shows the hydrogen production at different initial temperatures. **At 30 °C and 33°C in dark fermentation system, 24 and 27 °C in photo fermentation system, the hydrogen production is affected by temperature fluctuations, and is greater than that of constant temperature system at the same initial temperature.** The opposite situation occurs when the initial temperature is 36 and 39°Cin **dark fermentation system**, 30 and 33°C in **photo fermentation system**. The reason may be due to the activity of the enzyme is negatively affected by the temperature increasing. The maximum hydrogen production occurs at 27°C while the minimum occurs at 24°C for photo fermentation system.

Point 14: Page 19, Table 3: The costs should also be provide in US\$ to make the findings more comparable.

Response: Thank you very much for your comments. We have revised Table3, changes are marked in red.

When the (net present value) NPV>0, it means the plan is economically feasible. Tp is the investment payback period, IRR is the internal return of rate. When IRR is higher than the benchmark discount rate (8%), the project has a favorable economic effect.³¹ According to market price at project site, the sale price of hydrogen is set 56 CHY/kg(7.65\$/kg). Based on the data of Table 3, the financial net present value (NPV) of dark and photo-fermentation system is 584,400 CNY(79,843\$/kg), the investment payback periods are estimated to be 6.86 years and IRR is 16.84% for dark- and photo-fermentation system(Supplementary Table 4), the payback period is lower than 10.28 years for hydrogen production with water electrolysis (IRR 10.28%)⁵⁵.

Table 3 Capital investment and cost analysis

Initial investment (10 ⁴ CNY)	Raw materials (t)	Reagent consumption (CNY/t H ₂)	Water (CNY/t H ₂)	Construction cost (CNY/t H ₂)	Maintenance cost (CNY/year)	Welfare (CNY/month)	Cost (CNY/t H ₂)
800, (109.33\$)	40	9,697.6 (1,325.35\$)	3,520 (481.07\$D)	950 (129.83\$)	12,000 (1,640.02\$)	3,000 (410\$)	43,875.16 (5,996.33\$)

Point 15: Correct error “Midpiont”

Response: Thank you very much for your comments. We have revised the error. Changes are marked in red.

The life cycle data are collected mostly from China and supplemented with the database in Ecoivent 3.1 so that the assessment results can be more representative in China. ReCiPe 2016 **Midpoint (H)**, adopted in the life cycle assessment, incorporates 18 categories:

Point 16: Maybe it could be pointed out what part of the CO₂ eq emissions are physical emissions of biogenic carbon dioxide (that could be captured for HyBECCS approaches) in order to calculate the maximum negative emission potential NEP

Response: Thank you very much for your comments. The GWP comes from fossil (indirect), due to fossil was consumed to production of materials required for reactor

construction.

The reason may be due to the addition of chemical reagents during the fermentation stage. This provides a research direction for us to reduce GW from biological hydrogen production processes. Supplementary Table.2 shows the GW from different hydrogen production technology. The GW of the hydrogen production via biomass gasification and coal gasification is approximately 10.56 kg CO₂ eq/kg H₂⁴⁵ and 18 kg CO₂ eq/kg⁴⁶, respectively. The GW of green hydrogen production via water electrolysis using wind power or solar power shows lower value varying from 9.4 to 0.3 kg CO₂ eq/kg^{47,48}. But renewable power generation is limited by land area available for photovoltaic panels and/or wind turbines, when the grid electricity is used to compensate for insufficient wind or solar electricity, the GW could reach to 25.93 CO₂ eq/kg H₂⁴⁹. Comparing to green hydrogen production via water electrolysis, biological fermentation hydrogen production system exhibits advantages in resource recycling and simultaneously achieving waste treatment and clean energy production. With the application of carbon capture and storage technology, the preparation process of materials gradually becomes cleaner, which will help reduce the GW of the biological fermentation hydrogen production system.

Through literature comparison, it was also found that different biomass conversion technologies exhibit different emission reduction capabilities based on LCA⁵⁰⁻⁵², caused by the different in assessment methods, functional units and system boundaries⁵³, but the final results all indicate development and utilization of biomass helps to net-zero emissions.

Point 17: 447/448: please check the proper usage of italics (italics only for strains, other words: normal)

Response: Thank you very much for your comments. We have checked the proper usage of italics, changes are marked in red.

The HAU-M1 was composed of oval-shaped *Rhodospirillum rubrum* (27%), rod-shaped *Rhodopseudomonas palustris* (28%), ellipsoid-shaped or short rod-shaped *Rhodopseudomonas capsulata* (25%), *Rhodobacter sphaeroides* (9%) and *Rhodobacter capsulatus* (11%)³⁸.

Point 18: scanned

Response: Thank you very much for careful review. We have revised.

Point 19: Spectrophotometer from which company? Path-length of cuvette?

Response: Thank you very much for your comments. **Agilent8453 UV-VIS spectrophotometer(Agilent, USA) with a scanning range of 190-1100 nm** was used to detect the absorption of light by photosynthetic bacteria. Path-length of cuvette is 10mm. The contents have been added, marked in red.

The logarithmic hydrogen producing bacteria are suspended in a 60% sucrose solution after centrifugation and washing, **scanned** with an **Agilent8453 UV-VIS spectrophotometer(Agilent, USA) with a scanning range of 190-1100 nm** to detect the absorption of light by photosynthetic bacteria⁵⁷.

Point 20: Filters from which company

Response: Thank you very much for your comments .The filters were purchased from Nantong Yinxing Optics Co., Ltd, China, the relevant information has been added.

Using the light from the xenon lamp that is closest to the solar spectrum as the light source, a filter device is installed in front of the light collector to obtain spectra of different wavelengths. The filters used in this article are 100 nm bandpass filters DTB400, DTB470, DTB540, DTB600, DTB700, **and DTB720(Nantong Yinxing Optics Co., Ltd, China)** to process the collected xenon lamp light, in order to ensure the uniformity of photosynthetic bacteria receiving light and avoid losses caused by light scattering, the xenon light is collected by a collector and transmitted to the glass bottle by optical filters(the reaction device is shown in Supplementary Figure 2a).

Point 20: LEDs from which company? Spectra? Intensities?

Response: Thank you very much for your comments. The LEDs used in the paper were purchased from Cree Shanghai Opto Development Limited. During experiments, the light intensity of LED is set to 3000Lx.

Point21: A short description about how the dry-weight determination and the hydrogen measurements were performed should be provided.

Response: Thank you very much for your comments. The method of the dry-weight determination and the hydrogen measurements have been added, marked in red.

The test conditions are set as follows: 25% of the inoculation amount, 30 °C of temperature, 3000 Lx of light, pH 7, and 10 g/L of glucose concentration.

The dry cell weight in culture medium by centrifuging of suspension for 5 min at 10,000 rpm , using distilled water to washed the sediment twice , and then drying it at 105 C° until constant. The gas chromatograph (Agilent, 6820 GC-14B) was used to determine the hydrogen concentration. The light intensity was measured with a digital Lux meter (TES-1330A, Taiwan, China). An oxidation–reduction potentiometer (SX712, Shanghai, China) was utilized to measure the oxidation–reduction potential (ORP) of the reaction solution. The pH was measured by a pH meter (PHS-3C, Shanghai, China).²⁶

Point 22: please check this sentence for correct English

Response: Thank you very much for your comments. We apologize the incorrect English of our manuscript. We invited native English teachers to revise the manuscript, changes were marked in red.

Point 23: Please avoid repetitions throughout the document (e.g. 536 -550)

Response: Thank you very much for your comments. We have revised the content, marked in red.

Life cycle assessment method: The system boundary can be divided into three stages:

1. raw material pulverization; 2. enzymolysis pretreatment; 3. fermentation based hydrogen production. According to ISO 14040 and ISO 14044, LCA for hydrogen production systems is made by using SimaPro 8.5.

REVIEWER COMMENTS

Reviewer #6 (Remarks to the Author):

Comments:

1. The authors seem to have duly addressed the comments from the reviewer 4. On the issues with wavelengths concerning the photosynthetic activity of purple bacteria, it is agreed that though there are certain advantages of using IR wavelengths spectrum for PPB growth as it seems to avoid algal growth, the authors seem to have worked on white LED lights which mostly emit light intensities in the visible range. I suggest the authors to add a discussion in this regard.
2. There are several biohydrogen production routes, I believe the title of the paper should reflect the "dark-photo fermentation" routes which was the focus of the study.

[Response to points raised by Reviewer#6]:

Point 1: The authors seem to have duly addressed the comments from the reviewer 4. On the issues with wavelengths concerning the photosynthetic activity of purple bacteria, it is agreed that though there are certain advantages of using IR wavelengths spectrum for PPB growth as it seems to avoid algal growth, the authors seem to have worked on white LED lights which mostly emit light intensities in the visible range. I suggest the authors to add a discussion in this regard.

Response: We thank you very much for your suggestion. Related discussions have been added. All changes were marked in red.

The growth and hydrogen production characteristics of photosynthetic bacteria under different light sources(light-emitting-diodes(LEDs) purchased from Cree Shanghai Opto Development Limited) are shown in Fig. 2c. **Photosynthetic bacteria show better growth and hydrogen production characteristics under yellow and blue light; the corresponding hydrogen production is 4.24 mol H₂/mol glucose and 3.74 mol H₂/mol glucose, respectively, which is higher than that of white light sources (LED). The reason might be due to the light emitted by white light sources is a composite light composed of multiple monochromatic lights, higher radiation energy is absorbed by photosynthetic bacteria, resulting in “light saturation effect” that reduces the activity of photosynthetic bacteria.**

Point 2: There are several biohydrogen production routes, I believe the title of the paper should reflect the “dark-photo fermentation” routes which was the focus of the study.

Response: We thank you very much for your suggestion and the opportunity to improve our manuscript. Yes, there are several biohydrogen production routes, dark-

photo fermentation is one of the routes. We have revised the title based on your suggestions. All changes were marked in red.

Towards Sustainable Biohydrogen Production: **Dark and Photo-Fermentation
Pilot-Scale Systems and Evaluation for Commercial Viability**

REVIEWERS' COMMENTS

Reviewer #6 (Remarks to the Author):

Thank you for revising the manuscript considering the comments.